# Mendelian randomization uncovers a protective effect of interleukin-1 receptor antagonist on kidney function

Jeong Min Cho[1], Jung Hun Koh[1], Seong Geun Kim[2], Soojin Lee[3,4], Yaerim Kim[5], Semin Cho[6], Kwangsoo Kim[7], Yong Chul Kim[1,4], Seung Seok Han[1,4,8], Hajeong Lee[1,4], Jung Pyo Lee[4,8,9], Kwon Wook Joo[1,4,8], Chun Soo Lim[4,8,9], Yon Su Kim[1,4,8,10], Dong Ki Kim [1,4,8] & Sehoon Park [1✉]

Interleukins (ILs), key cytokine family of inflammatory response, are closely associated with kidney function. However, the causal effect of various ILs on kidney function needs further investigation. Here we show two-sample summary-level Mendelian randomization (MR) analysis that examined the causality between serum IL levels and kidney function. Genetic variants with strong association with serum IL levels were obtained from a previous genome-wide association study meta-analysis. Summary-level data for estimated glomerular filtration rate (eGFR) were obtained from CKDGen database. As a main MR analysis, multiplicative random-effects inverse-variance weighted method was performed. Pleiotropy-robust MR analysis, including MR-Egger with bootstrapped error and weighted median methods, were also implemented. We tested the causal estimates from nine ILs on eGFR traits. Among the results, higher genetically predicted serum IL-1 receptor antagonist level was significantly associated with higher eGFR values in the meta-analysis of CKDGen and the UK Biobank data. In addition, the result was consistent towards eGFR decline phenotype of the outcome database. Otherwise, nonsignificant association was identified between other genetically predicted ILs and eGFR outcome. These findings support the clinical importance of IL-1 receptor antagonist-associated pathway in relation to kidney function in the general individuals, particularly highlighting the importance of IL-1 receptor antagonist.

[1] Department of Internal Medicine, Seoul National University Hospital, Seoul, Korea. [2] Department of Internal Medicine, Inje University Sanggye Paik Hospital, Seoul, Korea. [3] Department of Internal Medicine, Uijeongbu Eulji University Medical Center, Seoul, Korea. [4] Department of Internal Medicine, Seoul National University College of Medicine, Seoul, Korea. [5] Department of Internal Medicine, Keimyung University School of Medicine, Daegu, Korea. [6] Department of Internal Medicine, Chung-Ang University Gwangmyeong Hospital, Gyeonggi-do, Korea. [7] Transdisciplinary Department of Medicine & Advanced Technology, Seoul National University Hospital, Seoul, Korea. [8] Kidney Research Institute, Seoul National University, Seoul, Korea. [9] Department of Internal Medicine, Seoul National University Boramae Medical Center, Seoul, Korea. [10] Department of Biomedical Sciences, Seoul National University College of Medicine, Seoul, Korea. ✉email: mailofsehoon@gmail.com

The kidneys are vital organs that contribute to various biological processes. Various factors control kidney function, including cytokines[1], neurohormonal responses[2], and vascular regulations[3]. The substantial socioeconomic burden of kidney dysfunction[4] warrants investigating the potential therapeutic or preventive effects of each factor on the impairment of kidney function.

Among the diverse regulators of glomerular filtration rates, inflammatory cytokines, including interleukins (ILs), directly affect kidney function[5,6]. Since the development and progression of major kidney diseases are linked to inflammation, the effect of these cytokines on kidney function has been shown in certain disease conditions. For instance, IL-1 and IL-6 are associated with kidney fibrosis and the decline of kidney function[7,8], and IL-18 has been proposed to have detrimental effects on ischemic kidney diseases[9,10]. Several types of ILs are targeted as immunomodulating agents, including a recombinant IL-1 receptor antagonist (IL-1ra), which demonstrated rapid and sustained reduction of the inflammatory response in inflammation-mediated organ dysfunction (e.g., heart failure and stroke)[11].

Mendelian randomization (MR) analysis is a method to estimate the causal effect of exposure on complex traits, overcoming the limitations of conventional observational studies by being less likely to be affected by confounding and reverse causation[12]. Furthermore, MR studies can provide valuable evidence to support the appropriate use of drugs in chronic kidney disease (CKD) patients. A previous MR study that did not identify a significant association between IL-6 inhibition and kidney function suggested MR evidence support the safe administration of tocilizumab, an IL-6 inhibitor, in patients with renal impairment[13]. Quantitative trait locus (QTL) studies enabled the discovery of genetic variants that affect a quantitative change of a particular trait, including the expression of one or more genes (eQTL) or protein (pQTL)[14,15]. Performing MR analysis with QTL instruments confers insight into identifying candidate genes and molecular targets for diseases by demonstrating the causal effect between gene and protein expression on traits[16–18].

In this study, we performed MR analysis to examine the causal effects of certain ILs on kidney function. We selected genetic instruments that satisfied two criteria (cis-pQTL and cis-eQTL) and were robustly associated with serum IL levels. We tested the causal estimates in multiple population-scale genetic datasets of kidney function traits, hypothesizing that certain IL-associated molecules would significantly affect kidney function.

## Results

**Characteristics of the data sources**. We performed a two-sample summary-level MR analysis of the serum IL levels and kidney function traits. The study flow diagram is presented in Fig. 1. The summary statistics of kidney function were obtained from the UK Biobank (UKB) and CKDGen genome-wide association study (GWAS) meta-analysis[19–21]. The mean age of the UKB participants was 57.6 ± 8.5 years and ~6.2% of the participants had CKD[22]. The median age of CKDGen GWAS meta-analysis participants was 54 years, and the median of mean estimated glomerular filtration rate (eGFR) values were 89 (interquartile range 81, 94) ml/min/1.73 m² [21]. In the GWAS meta-analysis of eGFR decline, the median annual eGFR decline across studies was 1.32 ml/min/1.73 m² per year[19].

**MR analysis of serum IL-1ra levels on kidney function**. Among the ILs, genetically predicted serum IL-1ra levels showed a consistently significant association ($P < 0.05$) with eGFR, which was also supported by the significant ($P < 0.05$) pleiotropy-robust MR results (Table 1). Using 20 single nucleotide polymorphisms

(SNPs; 18 pQTL and two eQTL), higher serum IL-1ra levels were significantly associated ($P < 0.05$) with higher eGFR in the multiplicative random-effects inverse-variance weighted (IVW) analysis (Fig. 2). In the MR analysis with log-transformed creatinine-based eGFR outcome statistics provided by CKDGen and UKB datasets, the eGFR increased by 0.28 % (standard error [SE], 0.11 %; $P$ value, 0.009) as genetically predicted standard deviation increase in serum IL-1ra concentration. The association was also consistent when the MR analyses were performed against the other two eGFR outcome statistics: creatinine-based and cystatin-C-based eGFR. Pleiotropy-robust MR sensitivity analyses, including MR-Egger with bootstrapped error and the weighted median method, supported these findings. The nonsignificant $P$ value of the intercept in MR-Egger indicated the absence of pleiotropic effects in the above findings. Furthermore, when we applied MR analysis toward the degree of annual eGFR decline, the result showed a significant finding, indicating that genetically predicted higher IL-1ra levels were significantly associated ($P < 0.05$) with a lower degree of annual eGFR decline. When we performed MR analysis with annual eGFR decline summary statistics from CKDGen and UKB datasets, the annual eGFR decline decreased by 2.18 % (SE, 1.09 %; $P$ value, 0.043) a genetically predicted standard deviation increase in serum IL-1ra concentration. The leave-one-out analysis results demonstrate the absence of a notable outlier effect in the calculated causal estimates (Fig. 3).

**MR analysis of other serum IL levels on kidney function**. The results of the summary-level MR analyses of the eight types of ILs (except IL-1ra) are summarized in Supplementary Table 1. There was a significant association ($P < 0.05$) between genetically predicted higher serum IL-2ra levels and higher eGFR in the multiplicative random-effects IVW model; however, this association was not replicated by pleiotropy-robust MR analyses (Fig. 2). MR analyses did not reveal causal association between genetically predicted serum IL levels (for IL-1α, IL-16, and IL-18) and eGFR. We performed a ratio of coefficient analysis with IL-6, IL-7, and IL-12p70, each provided with a single genetic instrument; however, no association between the serum IL concentrations with a single instrumented variant and eGFR were observed. The MR estimate for IL-6 was calculated using single genetic instrument (rs7808457), because rs57349960 was not included in the outcome GWAS database.

**Sensitivity analysis of serum IL levels on kidney function**. In the sensitivity analysis excluding cis-eQTLs (Supplementary Table 2) and palindromic SNPs (Supplementary Table 3), we found that the MR estimates for genetically predicted IL-1ra levels were consistent with the main analysis, which revealed significant associations with eGFR and eGFR decline.

## Discussion

In this MR study, we investigated the causal association between genetically predicted serum IL concentrations and eGFR, using cis-QTL instruments and GWAS summary statistics. These results demonstrate a significant causal effect of serum IL-1ra levels on kidney function. Higher genetically predicted serum IL-1ra levels were significantly associated with a higher eGFR in the two population-scale genetic datasets, consistent with the results of the main MR analysis and various MR sensitivity analyses. Genetically predicted serum IL-1ra levels were also associated with the degree of annual eGFR decline, suggesting that higher IL-1ra levels ameliorate the rate of eGFR decline.

Because many kidney diseases are related to immune responses, understanding the mechanisms of circulating cytokines,

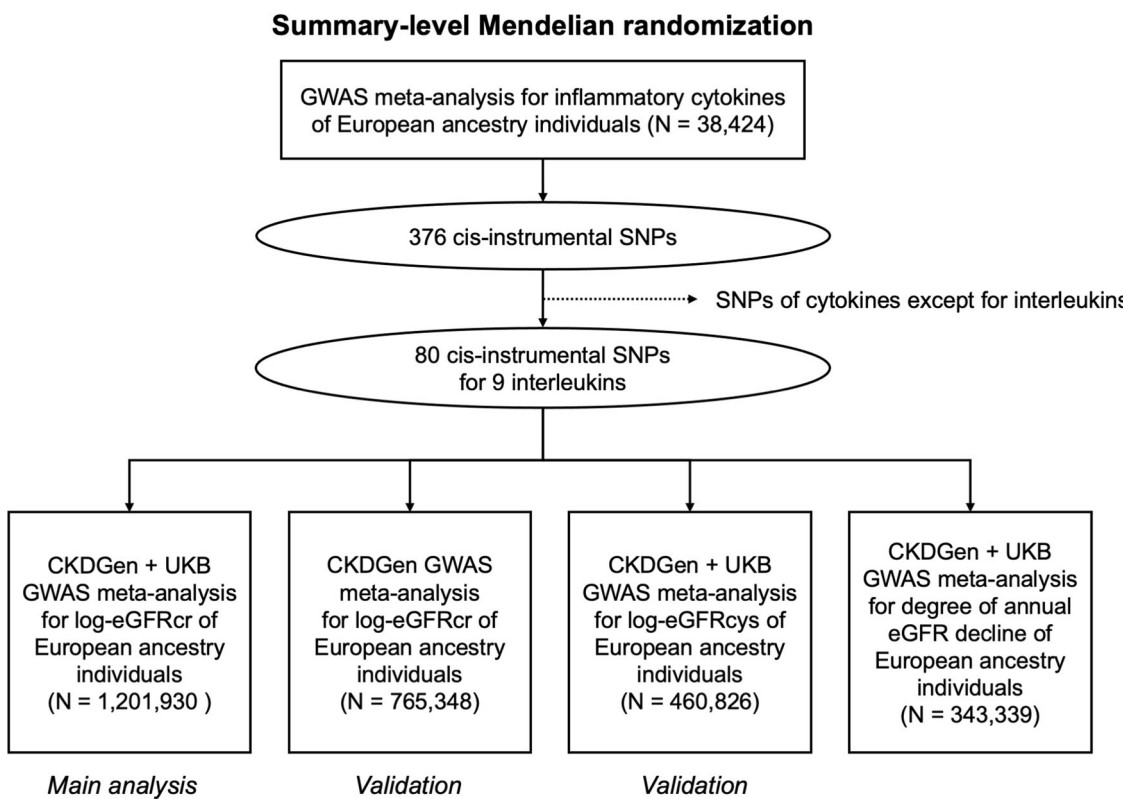

**Summary-level Mendelian randomization**

**Fig. 1 Study flow diagram.** The study included summary-level Mendelian randomization analysis testing the causal effects between circulating interleukin (IL) level and kidney function. The cis-genetic instruments for genetically predicted serum IL levels, including cis-pQTL and cis-eQTL instruments, were developed from the previous genome-wide association study (GWAS) meta-analysis including 38,424 participants. Among 376 SNPs, 80 SNPs of nine ILs were utilized for the Mendelian randomization analysis. The summary statistics for kidney function traits were obtained from four respective GWAS meta-analysis datasets: the meta-analysis for creatinine-based log-eGFR values of CKDGen and UKB ($n = 1,201,930$), creatinine-based log-eGFR values from the phase 4 CKDGen study ($n = 765,348$), cystatin-C-based log-eGFR values from CKDGen and UKB ($n = 460,826$), and degree of annual eGFR decline from CKDGen and UKB ($n = 343,339$). Below the boxes of each dataset, it is indicated whether the datasets were analyzed as "Main analysis" or "Validation". UKB UK Biobank, SNP single nucleotide polymorphism, GWAS genome-wide association study, eGFRcr estimated glomerular filtration rate by creatinine, eGFRcys estimated glomerular filtration rate by cystatin-C.

**Table 1 Positive finding of summary-level MR analysis of genetically predicted serum IL levels using cis-pQTL and cis-eQTL instruments with CKDGen data (IL-1ra).**

| Interleukin | [a]Outcome | MR-Egger intercept P | MR methods | eGFR change beta (%) | Standard error (%) | P value |
|---|---|---|---|---|---|---|
| IL-1ra | 1) Creatinine-based log-eGFR values (CKDGen + UKB) | 0.16 | MR-IVW | 0.28 | 0.11 | 0.009 |
| | | | MR-Egger | 0.36 | 0.12 | 0.003 |
| | | | Weighted median | 0.32 | 0.11 | 0.004 |
| | 2) Creatinine-based log-eGFR values (CKDGen) | 0.53 | MR-IVW | 0.32 | 0.09 | 3E-04 |
| | | | MR-Egger | 0.25 | 0.14 | 0.04 |
| | | | Weighted median | 0.36 | 0.13 | 0.005 |
| | 3) Cystatin-C-based log-eGFR values (CKDGen + UKB) | 0.16 | MR-IVW | 0.35 | 0.17 | 0.039 |
| | | | MR-Egger | 0.73 | 0.21 | 0.001 |
| | | | Weighted median | 0.52 | 0.18 | 0.003 |
| | 4) Degree of annual eGFR decline (CKDGen + UKB) | 0.29 | MR-IVW | −2.18 | 1.09 | 0.043 |
| | | | MR-Egger | −2.42 | 1.71 | 0.078 |
| | | | Weighted median | −2.85 | 1.48 | 0.049 |

*MR* Mendelian randomization, *eGFR* estimated glomerular filtration rate, *MR-IVW* multiplicative random-effects inverse-variance weighted.
Both cis-pQTL and cis-eQTL instruments were included as genetic instruments of genetically predicted serum IL-1ra level.
[a]Four summary statistics for kidney function traits were used for MR analyses: (from the first row) (1) creatinine-based log-eGFR values of the CKDGen and the UKB data[20], (2) creatinine-based log-eGFR values from the phase 4 CKDGen study[21], (3) cystatin-C-based log-eGFR values from the CKDGen and UKB[20], and (4) degree of annual eGFR decline, including CKDGen and UKB[19].
All effect sizes were aligned and scaled towards genetically predicted standard deviation increase in serum IL concentration for % change in eGFR.

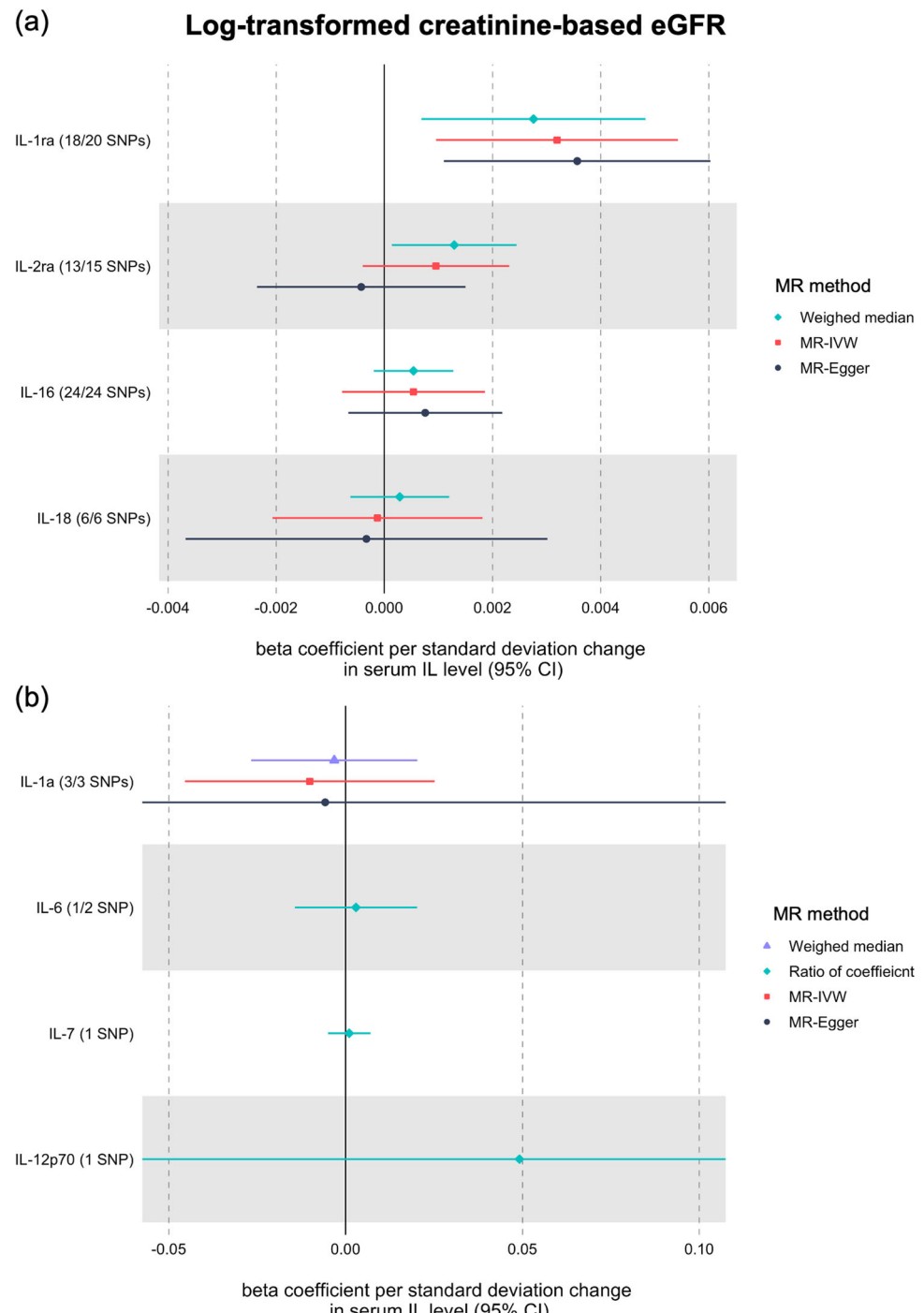

**Fig. 2 Two-sample Mendelian randomization analysis results of the causal effects of nine types of serum interleukin levels on kidney function.** The causal estimates were derived from the UK Biobank and CKDGen genome-wide association study meta-analysis for log-transformed creatinine-based eGFR. The shapes indicate beta coefficients per standard deviation change in corresponding serum interleukin level, and error bars indicate the 95% confidence interval. **a** The rhombus, square, and circle shapes indicate estimates calculated from weighted median, MR-IVW, and MR-Egger, respectively. **b** The triangle, rhombus, square, and circle shapes indicate estimates calculated from the weighted median, ratio of coefficient, MR-IVW, and MR-Egger, respectively. eGFR estimated glomerular filtration rate, IL interleukin, 95% CI 95% confidence interval, MR-IVW multiplicative random-effects inverse-variance weighted.

including ILs, and their effect on the kidney microenvironment is crucial for preventing and treating kidney diseases[23]. The association between IL profiles and kidney function has been suggested in previous cross-sectional observational studies[5,6]. Moreover, several experimental and clinical studies are being conducted to find molecular targets that have immune-

modulating effects in patients with kidney disease[8–10,24]. However, causality could not be inferred due to methodological limitations, and the roles of ILs related to general kidney function are still not fully understood. Our study found MR evidence of the protective effect of IL-1ra, a naturally occurring IL-1 receptor antagonist, demonstrating a significant association with eGFR.

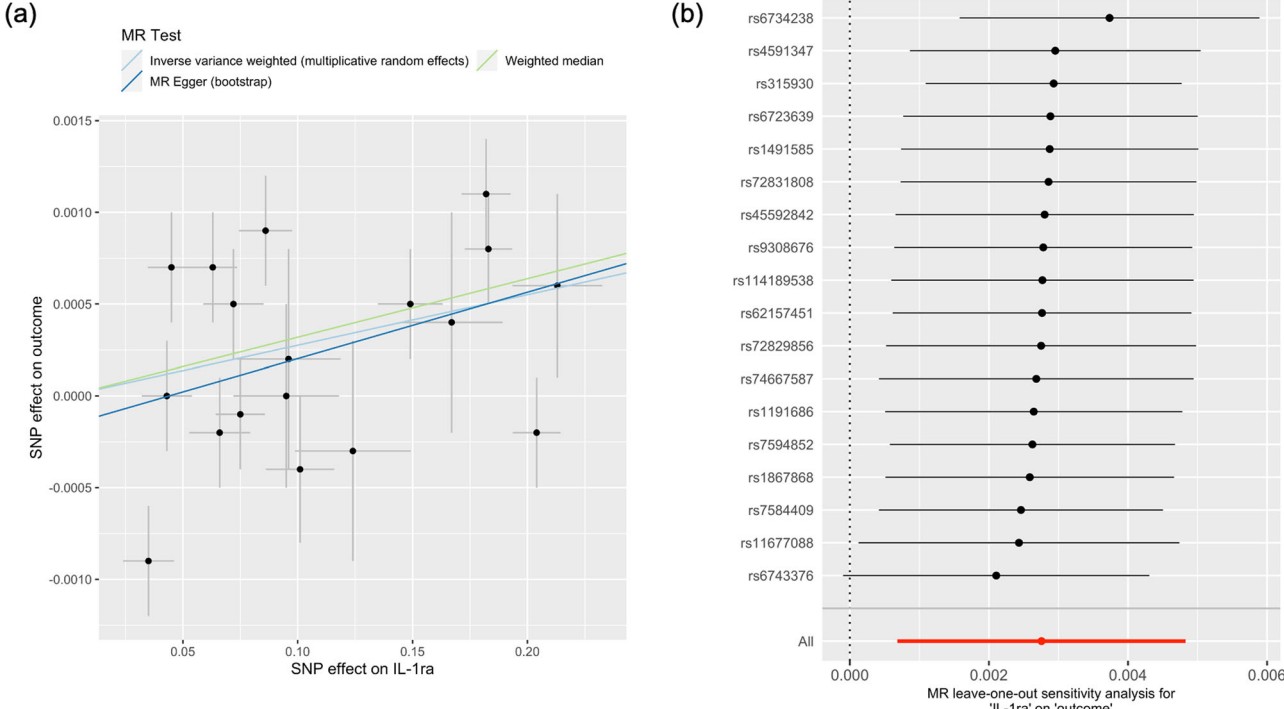

**Fig. 3 Mendelian randomization plots of the causal effect of serum interleukin-1 receptor antagonist (IL-1ra) level on kidney function. a** Scatter plot of the effect sizes of the SNP-serum IL-1ra level association (x axis, SD units) against the SNP-log-eGFRcr associations (y axis, beta) with standard error bars. The slopes of the lines correspond to the causal estimates per method. **b** Leave-one-out sensitivity analysis. Each black point represents the MR-IVW method applied to estimate the causal effect of serum IL-1ra level on kidney function excluding that certain variant from the analysis. The red point indicates the IVW estimate using all SNPs. There are no instances where the exclusion of a single SNP leads to dramatic changes in the overall result. Outcome statistics were obtained from the UK Biobank and CKDGen genome-wide association study meta-analysis of log-eGFRcr. MR Mendelian randomization, IL interleukin, SNP single nucleotide polymorphism.

Our study has several strengths. First, we performed MR analysis, which is less affected by bias from confounding effects or reverse causation and can provide evidence of a causal association between genetic predisposition to IL levels and kidney function. Second, we integrated the cis-pQTL and cis-eQTL information of ILs with summary-level outcome data from GWAS in the MR framework to estimate the causal effect of serum IL levels and the corresponding gene expression on kidney function traits. This further increased the power of MR estimates in identifying trait-associated loci, as well as reducing horizontal pleiotropy[25,26]. As genetic variants encoding a drug target have a similar effect as pharmacological agents that target this gene location, our findings may suggest a potential drug target for modifying kidney diseases[27]. Third, various types of ILs that are known to be associated with the pathogenesis of kidney diseases were included in the study[5,6]. Fourth, the MR results of IL-1ra were consistently validated even when we analyzed the association between IL-1ra and eGFR using four different outcome statistics on kidney traits, providing evidence for the protective effect of serum IL-1ra levels on kidney function.

IL-1 is a key pro-inflammatory cytokine that activates immune cells and promotes secretion of downstream cytokines including IL-6 and TNF-a, further driving amplification of innate immunity and inflammation in various organs and tissues[28–30]. Currently, drugs targeting IL-1, including recombinant IL-1 receptor antagonist (anakinra), monoclonal IL-1β antibodies (canakinumab), and IL-1 traps (rilonacept) are approved for the treatment of inflammation-mediated diseases, including arthritis, gout, type 2 diabetes, and heart failure[11]. In the kidney, activated IL-1 pathway causes cell stress, tissue damage, and fibrosis, eventually leading to loss of kidney function[8]. Kidney dysfunction may also

be contributed by systemic release of IL-1 in lupus or diabetes that promotes leukocyte adhesion and vascular leakage in the glomeruli[31]. Experimental evidence suggests anti-hypertensive, anti-fibrotic effects of IL-1 receptor blockade in kidney diseases[32,33]. Our observations are in line with the findings of these experimental studies, elucidating the important role of IL-1 pathway in kidney function impairment. Furthermore, in a clinical trial of IL-1 inhibitors that primarily investigated the effect of an IL-1β inhibitor for lowering the risk of hypertension and major adverse cardiovascular events, significant benefits for preventing adverse cardiovascular events were observed in a subgroup of patients with mild CKD[34]. The current study's findings provide MR evidence supporting the use of IL-1 blockade as a potential therapeutic target aimed at preserving kidney function or mitigating the adverse events associated with the deterioration of kidney function among individuals without pre-existing kidney function abnormalities.

Another valuable insight produced by implementing MR analysis between genetically proxied IL-1ra and kidney function, in addition to the identification of the underlying mechanisms of cytokine-associated inflammatory pathways and kidney function, is that the evidence for the safe administration of pharmacological agents in CKD patients could be provided by MR analysis. For example, in the previous MR study that investigated the association between genetically proxied IL-6 inhibition and kidney function, no significant causal association was found between IL-6 inhibition and kidney function. This finding supports that pharmacological IL-6 inhibition is unlikely to have a direct adverse effect on kidney function[13]. Given that patients with CKD are excluded from up to 75% of all randomized-controlled trials[35], the findings from the MR study can provide reliable

evidence for the appropriate use of IL inhibitors in CKD patients. According to studies on IL-1 inhibitors, including the CANTOS or IL-1ra Arthritis Study[36,37], no dose adjustment has been suggested for patients with renal impairment. However, it should be noted that this drug has not been studied in patients with moderate or severe kidney dysfunction. Further clinical trials are necessary to investigate the safety of IL-1 inhibitors in patients with impaired kidney function, and the current study suggests the possible safe administration of IL-1 inhibitors in CKD patients by providing MR evidence.

Our study has some limitations. First, the number of genetic instruments was insufficient for some ILs, which restricted the type of sensitivity analysis that could be performed. Although the strengths of the instruments were evaluated using F-statistics and proportion of the explained variance ($r^2$)[38], there were weak-powered instruments with F-statistics <10. It should be noted that there is potential for false-negative bias, particularly when the true effect is below the minimum effect size (Supplementary Table 4). Thus, one may not preclude the possibility of a kidney function effect based on the null findings in this study. Specifically, we could expect that the IL-1$\alpha$, which agonizes the IL-1 receptor, may produce an inverse effect on kidney function. However, the current study could not support the inverse association between serum IL-1$\alpha$ and kidney function. This may be explained by the broader biological function of IL-1ra than that of IL-1$\alpha$, including the blockade of IL-1 receptors from binding with IL-1 agonists[39]. However, the difference in the statistical power may have caused the discrepancy. Second, the effect estimates of MR analysis are for lifetime exposure to genetic predisposition and does not consider temporal and spatial fluctuations in gene expression in tissues. Additionally, as the overall effect sizes in the results were small, a transient change with a higher degree of ILs may have different effects on kidney function. Third, the generalizability is limited as the analysis was mainly based on European ancestry samples.

In conclusion, we examined the causal effects of various ILs on kidney function using MR analysis and found a positive association between genetically predicted IL-1ra levels and kidney function traits. Based on our MR results, IL-1ra may be highlighted for its effects on eGFR changes in the general population. However, further studies are required to ascertain the specific biological pathways involving ILs, particularly IL-1ra, in kidney function.

## Methods

**Ethics and inclusion statement**. This study was approved by the Institutional Review Board (IRB) of the Seoul National University Hospital (IRB number: E-2301-066-1394). The summary statistics of the kidney function traits are from the public domain (http://ckdgen.imbi.uni-freiburg.de/) of CKDGen GWAS meta-analysis database for kidney function traits. The requirement for informed consent was waived because the study analyzed anonymous public data and summary statistics.

**Study setting**. The current study is a two-sample summary-level MR analysis to examine the association between serum IL levels and kidney function. The genetic instruments for serum IL levels were developed from the previous study[40]. The outcome GWAS summary statistics for kidney function traits were obtained from the CKDGen consortium, which is widely used for MR analysis towards kidney function traits[41–43]. As a main analysis, we assessed the causal effects of circulating concentrations of nine types of IL, including IL-1$\alpha$, IL-1ra, IL-2ra, IL-6, IL-7, IL-8, IL-12p70, IL-16, and IL-18, on kidney function using summary-level MR analyses. The examined ILs were previously reported to be associated with the pathogenesis of kidney inflammation, fibrosis, and injury[8,31,39,44–49]. Then, we also performed sensitivity analyses to ascertain the robustness of our findings[50].

**Genetic instrument for MR analysis**. Genetic instruments were developed from a previous GWAS meta-analysis study including genetic information of 47 inflammatory cytokines[40]. Using these genetic instruments, a total of 80 cis-instruments (60 as cis-pQTL and 20 as cis-eQTL) that are strongly associated with the effect of

circulating IL concentration were developed (Supplementary Tables 5, 6)[40]. Two cis-instrumental selection criteria were used to identify variants that reflected the effect of circulating cytokine levels and are described in Supplementary Note 1. To mitigate the potential loss of causal variants in a cis-region MR study due to a small correlation threshold, clumping was performed with a pairwise linkage disequilibrium threshold of $r^2 < 0.1$.

**Kidney function outcome in the MR analysis**. The CKDGen consortium provides a publicly available database for GWAS and meta-analyses of kidney function (https://ckdgen.imbi.uni-freiburg.de). Because the genetic instruments were developed from individuals of European ancestry in the SCALLOP consortium and the Finnish population, we used the summary statistics of individuals of European ancestry and used the data as the outcome statistics in our two-sample MR analysis. We used two independent datasets, the CKDGen consortium data, and the UKB, as outcome datasets for log-transformed eGFR values because validation analysis is crucial for the validity of a genetic study.

We used three eGFR outcome datasets as eGFR values are a current standard parameter for the assessment of kidney function; (1) the meta-analysis for creatinine-based log-eGFR values of the CKDGen and UKB data from 2021 ($n = 1,201,930$) which has strength as this is the currently largest GWAS meta-analysis for eGFR trait[20], (2) creatinine-based log-eGFR values from the phase 4 CKDGen study from 2019 ($n = 765,348$)[21], and (3) cystatin-C-based log-eGFR values from the CKDGen and UKB ($n = 460,826$) from 2021[20], which has particular strength in that the dataset can be used to test the causal estimates towards cystatin-C-based eGFR values which is less affected from non-kidney factors than creatinine-based levels[51]. The main analysis was performed using the largest GWAS meta-analysis of creatinine-based log-eGFR from CKDGen and UKB because of the greater statistical power and lower possibility of weak-instrumental bias. Nevertheless, considering the potential for healthy volunteer bias resulting from the relatively lower prevalence of CKD in the UKB dataset[22], validation analyses employing the other two GWAS meta-analysis datasets were conducted. Additionally, the UKB dataset was independent of the CKDGen data, fulfilling an independent validation analysis, and was independent of the samples included in the GWAS for genetic instrument development. Such a two-sample MR without participant overlap has statistical power even if the results are affected by instrumental power because the potential bias would be towards false-negative findings. Therefore, a positive finding from such an independent two-sample MR is more likely to reflect true causal effects[52].

Furthermore, to test whether IL levels have causal effects on accelerated eGFR decline rather than on a static value, we implemented MR analysis using a GWAS meta-analysis of the degree of annual eGFR decline from CKDGen and UKB from 2022 ($n = 343,339$) as another outcome dataset[19].

**MR assumptions**. In the context of selecting the genetic instruments for circulating IL concentration and performing MR analysis, we considered the three assumptions of MR analysis to demonstrate valid causal effects[53]. First, the genetic instrument is robustly associated with the exposure of interest (the relevance assumption), second, the genetic instrument is independent of the confounding factors of the association between the exposure and the outcome of interest (the independence assumption), and third, the genetic effects occur only through its association with the exposure of interest (the exclusion-restriction assumption).

For the relevance assumption, genetic instruments provided from the previous meta-analysis results of the Finnish and SCALLOP GWAS were strongly associated with circulating cytokine concentrations at genome-wide significance ($P < 5 \times 10^{-8}$) divided by the number of cytokines analyzed[40,53,54]. In addition, the F-statistic and $r^2$ for each genetic variant based on the circulating protein concentration were calculated as a measure of the strength of the instrument[40,53–55].

Horizontal pleiotropy occurs when a genetic variant affects the outcome outside the exposure pathway[56]. For the independence and exclusion-restriction assumption, both of which require the absence of horizontal pleiotropy and possible confounder phenotype, pleiotropy-robust MR analysis methods were performed as sensitivity analysis[50]. The directions of the variants' causal effects were ascertained using MR Steiger filtering to establish that there is no direct effect of the genetic instruments on kidney function traits, which may potentially violate the exclusion-restriction assumption[57]. Steiger filtering results for each instrumental variant demonstrated evidence of causality in the expected direction.

**Statistics and reproducibility**. Two-sample summary-level MR analysis was performed using two sets of genetic instruments (cis-pQTLs and cis-eQTLs) for each IL on the four outcome datasets. First, we implemented a multiplicative random-effects IVW method when more than one genetic instrument was available to construct an instrumental variable for a given IL. Provided with only a single genetic instrument, the ratio of coefficient, also referred to as the Wald ratio, was applied to measure MR estimates[58]. Multiplicative random-effects model is the current standard for main MR analysis as the method allows balanced pleiotropic effects[50,59]. In addition, leave-one-out analysis, leaving one SNP out of the analysis at a time, was performed to inspect whether there was bias from an extreme outlier suspected with pleiotropic effect. The MR estimates were converted into the degree

of change (percentage [SE]) of log-transformed eGFR according to the genetically predicted standard deviation in serum IL concentration to facilitate interpretation.

**Sensitivity analysis for summary-level MR analysis.** To explore the possibility of horizontal pleiotropy, pleiotropy-robust MR analyses that hypothesized the inclusion of pleiotropic variants were implemented as sensitivity analyses when ≥3 instrument variants were available. First, MR-Egger regression analysis with bootstrapped standard error provides a pleiotropy-robust causal estimate that is valid even if all genetic variants are invalid due to a violation of the independence assumption[60]. A significant MR-Egger intercept $P$ value would raise the suspicion of directional pleiotropy. Second, the weighted median method provides a valid causal estimate even up to 50% of the weighted instruments are invalid[61], and this method was implemented as another sensitivity MR analysis.

**Other statistical aspects of the MR analysis.** We performed two additional sensitivity MR analyses: (1) excluding cis-eQTL instruments and (2) excluding palindromic SNPs to test whether differences in instrument development methods or uncertain harmonization might have influenced the results. As cis-eQTL instruments are near the genomic location of the influenced genes that can influence the expression of multiple genes, potential pleiotropy due to cis-eQTL instruments can occur[62,63]. Therefore, sensitivity analysis with the exclusion of cis-eQTL instruments was implemented to assess the validity of the causal inference.

We considered the possibility of false-positive findings from multiple comparisons as the causal estimates were tested for multiple types of ILs. To address this issue, we performed multiple sensitivity analyses and only considered an MR result as a true positive signal when the significance threshold of two-sided $P$ was lower than 0.05, across all the sensitivity analyses performed per IL.

The minimum effect size of each IL for a power of 80% was calculated in the application for calculating the post hoc power (https://shiny.cnsgenomics.com/mRnd/) using the meta-analysis dataset for log-eGFRcr of the CKDGen and UKB datasets and $r^2$ values of the instrumental variables of each IL at a type-I error rate of 0.25.

The summary-level MR analysis was performed using the "TwoSampleMR" package in R (version 0.4.26).

**Reporting summary.** Further information on research design is available in the Nature Portfolio Reporting Summary linked to this article.

## Data availability

All data used in this work are presented in the Supplementary information that accompanies the manuscript and is available in the original publications. The data used in this study is publicly available on the consortium website of The CKDGen (URL: https://ckdgen.imbi.uni-freiburg.de/)

## Code availability

This manuscript used public software those are available online. The names of the software are presented in the Methods and the detailed statistical codes can be found in the related resources.

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

## Acknowledgements
This work was supported by the National Research Foundation of Korea (NRF) grant funded by the Korea government (MSIT) (no. 2021R1A2C2094586). This work was also supported by the Research Fund from Seoul National University Hospital (3020220160).

## Author contributions
The corresponding author attests that all of the listed authors meet the authorship criteria and that no others meeting the criteria have been omitted. J.M.C., J.H.K., S.G.K., S.L., Y.K., S.C., Y.C.K., S.S.H., H.L., and J.P.L. performed the main statistical analysis including data curation, formal analysis, and investigation. K.K. contributed to the investigation and methodology. K.W.J., C.S.L., Y.S.K., D.K.K., and S.P. contributed to the conceptualization and design of the study. S.P. advised on statistical aspects and interpreted the data. D.K.K. and S.P. offer advice regarding data interpretation and supervision. S.P. obtained funding and supervised the overall project. All of the authors participated in drafting the manuscript. All of the authors reviewed the manuscript and approved the final version to be published.

## Competing interests
All authors declare no competing interests
