## [Peer Review File · Communications Biology]

Reviewers' comments:

Reviewer #1 (Remarks to the Author):

The authors have presented the eGFR data obtained from CKDGen and UK Biobank database. They have performed a two-sample summary-level Mendelian randomization (MR) analysis to investigate the causal relationships between the IL-associated molecules and kidney function using eGFR.

The methodology seems well written, and the authors explained all details in their statistical analysis and its assumptions.

The paper does provide new information regarding the link between renal function (eGFR) and IL-1ra levels.

Comments:

1- Did you consider gender in your study? If yes, is there any difference?

2- Did the authors consider any systemic disease, such as diabetes or heart failure, which can affect the GFR? Is there the exact link between the IL-1ra levels and eGFR in these diseases or other eGFR-affecting diseases? Please add your response to the discussion part.

3- It would be great if the authors added more explicit clinical implications of their findings in the discussion part. The discussion section should be completed with these explanations and the generalizability of the findings.

Reviewer #2 (Remarks to the Author):

In this manuscript, the authors conducted a two-sample MR between interleukins and eGFR in Europeans. MR studies on this topic are scarce and the choice of methods is fine. However, more care is needed in the reporting of methods, results, as well as the discussion of results. Please see below for my concerns:

Abstract

- Please state IL-1ra in full
- Suggest to limit the conclusion to IL-1ra, instead of "IL-1 associated pathway" because results with IL-1 alpha were mostly null

Introduction

- There is a previous MR study that looked at interleukin and renal functions (PMID: 33393675) and needs to be mentioned, as well as discussed in the discussion.
- MR can't minimise confounding and reverse causation, but effect estimates obtained using MR are less likely to be affected by confounding and reverse causation. Please amend

Method

- Since there are a lot of ILs, please explain how the authors arrived at the final list of ILs
- Suggest not to mention the component studies of meta-analysis to reduce confusion
- Please explain if clumping was involved when selecting genetic instruments and justify if not
- Please provide a reference for using the median of r^2 to depict strength of instruments. This method could possibly include weak instruments, particularly some instruments in the supplemental tables had F-stat below 10, implying results could be biased by weak instruments.
- Please clarify the source of exposure data. It was mentioned that the analysis was independently replicated using CKDGen and UKBiobank, please state which outcome was replicated as this was not shown in Figure 1.
- For creatinine-based log eGFR, I wonder what is the value added of using CKDGen alone alongside of the meta-analysis of CKDGEN + UKB, which is supposed to be of greater statistical power?

- For cystatin C-based log eGFR, it is stated in the method that the data was from UKB, but in Figure 1 it is shown as CKDGen + UKB. Please clarify the data source.
- Please provide reference for the sentence "cystatin C-based eGFR values which is less affected from non-kidney factors than creatinine-based levels".
- Please state clearly that GWAS of annual eGFR decline is a meta-analysis of data from CKDGen and UKB.
- Please explain specifically what was done to "ascertain the direction of the variants' effects" (P.9), as this sentence does not seem to be relevant to the next sentence.
- MR Egger could not provide a "consistent" estimate. Do the authors mean "valid"?
- The ability to calculate P of intercept in MR Egger is not a strength. Please delete
- Please explain how the exclusion of cis-eQTL instruments would improve the analysis
- Why weren't palindromic SNPs excluded in the main analysis, as done in default in the "TwoSampleMR" package?
- The authors stated that they considered the issue of multiple comparison but used a p-value threshold of 0.05 for statistical significance. Please further elaborate on this.

Result

- It was stated in the method that the eGFR values were logged in the outcome GWAS, but in the results the authors reported the change in eGFR as percentages. Please explain.
- Please state where the results of cystatin-C-based eGFR, as well as those of MR-Egger, were presented (which supp table?)
- Please explain how rs57349960 was selected to replace rs7808457 for IL-6.

Discussion

- Please discuss the possibility of drug use among participants in outcome GWAS affecting the eGFR measurements, thus affecting the results of the current study.
- The paragraph explaining IL-1 should appear earlier in the discussion, instead of behind the strengths.
- Please further elaborate what the authors meant by "the MR results of IL-1 alpha were not counterproductive...".
- Your result shown that horizontal pleiotropy was not evident through MR-Egger intercept. Why still list this as a limitation?

Reviewer #3 (Remarks to the Author):

In this study, the authors attempted to explore the relationship between Interleukin-receptor antagonist on kidney function through Mendelian randomization analysis. The subject of the study is very interesting, but there are some deficiencies in the choice of data and the use of methods.

- 1.The CKDGen database contains CKD, serum creatinine, uric acid protein, proteinuria and other data. Why did the authors choose only eGFR to measure renal function? And did the data used by the authors come from data from which year in the database? This should be explicitly described in the text.
- 2.What are the specific screening criteria for instrumental variables? What is their F-value?
- 3.Post hoc power for univariable primary analyses should be calculated to estimate the minimum effect size.
- 4.If only 1 instrument snp is available, the Wald ratio should be used.
- 5.There is a potential association between different IL-ras, and the authors should perform multivariable Mendelian randomization simultaneously to exclude potential effects.

Response Letter

Editorial Board: 1

You will see from their comments below that while they find your work of considerable interest, some important points are raised. We are interested in the possibility of publishing your study in Communications Biology, but would like to consider your response to these concerns in the form of a revised manuscript before we make a final decision on publication.

We therefore invite you to revise and resubmit your manuscript, taking into account the points raised. In particular, please provide more information regarding the selection of instruments, their strength and whether any sensitivity analyses are necessary to strengthen the findings.

Response: Thank you for the comment. We received very helpful comments from the reviewers requesting additional analyses that are worth considering in a robust Mendelian randomization study. The summary of the major requested analyses and the brief descriptions of the revisions made are listed below.

1. Clinical implications of the association between IL-1ra and kidney function

In the current study, the causal effects of various ILs on kidney function were investigated. Among the ILs, we found a significant association between genetically predicted serum IL-1ra levels and kidney function. IL-1 is a key inflammatory cytokine that negatively affects kidney function [1-4], and our findings suggest that IL-1ra has the potential to protect kidney function and may be a candidate for preventive cytokine targets for kidney function impairment. A clinical trial on IL-1 inhibitors showed a protective effect of IL-1 inhibitors on cardiovascular complications; however, the association with kidney outcome was not identifiable because the study was not powered enough [5-6]. In this regard, our study has the strength of providing MR evidence for the potential clinical use of IL-1 inhibitors to protect kidney function. This could be valuable data considering that patients with CKD are excluded from up to 75% of all randomized controlled trials, challenging the determination of medication dosage in these patients [8].

In the revised manuscript, we have reviewed the current clinical trials investigating the efficacy of IL-1 inhibitors and indicated a gap in our knowledge regarding the association between IL-1 inhibitors and kidney function. As a clinical implication, we suggest that IL-1ra may be a possible candidate for clinical trials targeting the protection of kidney function.

2. Methods to assess the strengths of instrumental variables and further explanation of some weak-powered instruments

The strengths of the instrumental variables were assessed using two statistics: F-statistics and proportion of variance explained (r^2). Assessment of the strength of the instruments by F-statistics or r^2 has been encouraged by the current guidelines for Mendelian randomization analysis, published by one of the leading researchers in the field of Mendelian randomization analysis field [9-10]. Current guidelines state that instrumental variables with an F-statistic <10 can be weak instruments. Please understand that we are very aware that some of our instrumental variables are weak-powered, with F-statistics of <10 . For example, the instruments of IL-1 α include F-statistics of 3.5 to 5.3, and some of the instruments of IL-16 were lower than 10.

In the revised manuscript, we have elaborated on the method to assess the strengths of instrumental variables and stated that weak-powered instruments can produce a possibility of false-negative bias. Nevertheless, we believe that the message of the study will be maintained despite some weak-powered instruments because the instrumental variables of IL-1 α , which is the main focus of our study, are sufficiently powered, presenting F-statistics >10 and $r^2 > 0.01$ in both eQTL and pQTL. Please also note that we were aware of the instrumental strengths and potential false-negative bias of the two-sample MR, which is clearly described in the Discussion section.

3. Clarification of various outcome data analyzed in the study

Because four meta-analyses were used as outcome summary data, we agree that clarification of various outcome data is needed. First, our main analysis was performed with a meta-analysis of CKDGen and UKB of log(eGFRcr) published in 2021 [11]. We then performed a replication analysis using two respective meta-analyses data: a meta-analysis of CKDGen of log(eGFRcr), 2019 [12] and a meta-analysis of CKDGen and UKB of log(eGFRcys), 2021 [11]. Lastly, we performed MR analysis using a meta-analysis of CKDGen and UKB for the degree of annual eGFR decline in 2022 [13].

In genetic analysis literature, it is common to perform various replication analyses to avoid false-positive bias. As MR causal estimates can be biased in a specific population or outcome measurement method, we aimed to test multiple outcome measures in certain populations. We believe the consistent finding in the CKDGen and UKB data (two datasets that contain the largest number of samples with kidney function measurement) is one of the most important strengths of our manuscript.

In the revised manuscript, we have clarified the source of each meta-analysis and the year of publication. In addition, we indicate whether each dataset was a main analysis or replication analysis in figure 1, below the boxes of each dataset. In each table, we numbered the meta-analyses and described the source of the outcome data in the table legend.

Figure 1 of the revised manuscript.

We truly thank the reviewers by raising very helpful and important comments for our study. The other detailed responses to the comments are attached below. We believe we made significant revisions addressing the points raised during this revision, and hope the current version would be more suitable for the publication process in *Communications Biology*.

Reference

- [1] Anders, H. J. Of Inflammasomes and Alarmins: IL-1 β and IL-1 α in Kidney Disease. *J Am Soc Nephrol* **27**, 2564-2575 (2016).
- [2] Netea, M. G., van de Veerdonk, F. L., van der Meer, J. W., Dinarello, C. A. & Joosten, L. A. Inflammasome-independent regulation of IL-1-family cytokines. *Annu Rev Immunol* **33**, 49-77 (2015).
- [3] Yazdi, A. S. & Drexler, S. K. Regulation of interleukin 1 α secretion by inflammasomes. *Ann Rheum Dis* **72 Suppl 2**, ii96-99 (2013).
- [4] Bandach, I., Segev, Y. & Landau, D. Experimental modulation of Interleukin 1 shows its key role in chronic kidney disease progression and anemia. *Sci Rep* **11**, 6288 (2021).

- [5] Rothman, A. M. *et al.* Effects of Interleukin-1 β Inhibition on Blood Pressure, Incident Hypertension, and Residual Inflammatory Risk: A Secondary Analysis of CANTOS. *Hypertension* **75**, 477-482 (2020).
- [6] Rothman AM, MacFadyen J, Thuren T, Webb A, Harrison DG, Guzik TJ, Libby P, Glynn RJ, Ridker PM. Effects of Interleukin-1 β Inhibition on Blood Pressure, Incident Hypertension, and Residual Inflammatory Risk: A Secondary Analysis of CANTOS. *Hypertension*. 2020 Feb;75(2):477-482. doi: 10.1161/HYPERTENSIONAHA.119.13642.
- [7] Ryan DK, Karhunen V, Walker DJ, Gill D. Inhibition of interleukin 6 signalling and renal function: A Mendelian randomization study. *Br J Clin Pharmacol*. 2021 Jul;87(7):3000-3013. doi: 10.1111/bcp.14725. Epub 2021 Feb 10. PMID: 33393675; PMCID: PMC8327328.
- [8] Charytan D, Kuntz RE. The exclusion of patients with chronic kidney disease from clinical trials in coronary artery disease. *Kidney Int*. 2006 Dec;70(11):2021-30. doi: 10.1038/sj.ki.5001934. Epub 2006 Oct 18. PMID: 17051142; PMCID: PMC2950017.
- [9] Burgess S, Davey Smith G, Davies NM, et al. Guidelines for performing Mendelian randomization investigations. *Wellcome Open Res*. 2020;4:186. Published 2020 Apr 28. doi:10.12688/wellcomeopenres.15555.2
- [10] Pierce BL, Ahsan H, Vanderweele TJ. Power and instrument strength requirements for Mendelian randomization studies using multiple genetic variants. *Int J Epidemiol*. 2011
- [11] Stanzick KJ et al. Discovery and prioritization of variants and genes for kidney function in >1.2 million individuals. *Nat Commun*. 2021 Jul 16;12(1):4350. doi: 10.1038/s41467-021-24491-0.
- [12] Wuttke, M. *et al.* A catalog of genetic loci associated with kidney function from analyses of a million individuals. *Nat Genet* **51**, 957-972 (2019). <https://doi.org/10.1038/s41588-019-0407-x>
- [13] Gorski, M. *et al.* Genetic loci and prioritization of genes for kidney function decline derived from a meta-analysis of 62 longitudinal genome-wide association studies. *Kidney Int* **102**, 624-639 (2022). <https://doi.org/10.1016/j.kint.2022.05.021>

Reviewer: 1

Remarks to the Author

The methodology seems well written, and the authors explained all details in their statistical analysis and its assumptions. The paper does provide new information regarding the link between renal function(eGFR) and IL-1ra levels.

Comment R1-1. Did you consider gender in your study? If yes, is there any difference?

Response R1-1. Thank you for your comments. We agree that considering gender differences may provide insight into the differences in kidney function according to serum IL levels by gender difference. However, please note that this study was a summary-level Mendelian randomization analysis using GWAS summary statistics of kidney function traits.

For outcome summary statistics of kidney function traits, we used 1) log(eGFRcr) GWAS meta-analysis of CKDGen and UK biobank for the main analysis and 2) log(eGFRcr) GWAS meta-analysis of CKDGen and 3) log(eGFRcys) GWAS meta-analysis of CKDGen and UK biobank for replication analysis. Additionally, 4) degree of annual eGFR decline meta-analysis of the CKDGen and UK biobanks was analyzed as summary statistics.

Please note that these studies did not provide sex-stratified summarized data. Specifically, each study in CKDGen conducted a GWAS for eGFRcr, adjusting for age, sex, and other study-specific covariates. In addition, the GWAS based on the UK biobank was conducted in a linear mixed model GWAS, including age, age², sex, age x sex, age² x sex, and 20 principal components as covariates, as recommended by the developers.

Please note that this is the general structure of common MR analyses, as most GWAS summary statistics provide results after sex adjustment. Thus, we believe that our results are in the appropriate standard for an MR study.

Comment R1-2. Did the authors consider any systemic disease, such as diabetes or heart failure, which can affect the GFR? Is there the exact link between the IL-1ra levels and eGFR in these diseases or other eGFR-affecting diseases? Please add your response to the discussion part.

Response R1-2. Thank you for your comments. Considering the effect of systemic diseases that can affect GFR and the linkage between IL-1ra levels and eGFR in individuals with certain systemic diseases may be an important issue. However, please understand that the implementation of MR analysis is predicated on the key assumption of a random allocation. Genetic variants used as instrumental variables in MR are randomly allocated during meiosis, and are thus not influenced by confounding factors (e.g., systemic diseases or comorbidities) that may be associated with the exposure and outcome of interest [1]. Therefore, systemic illness or comorbidity, which appears after birth, does not influence the random allocation of genetic variants at conception and is assumed to be already randomized when investigating the

exposure/outcome relationship in MR analysis. Additionally, when the linear mixed model GWAS for log(eGFRcr) and log(eGFRcys) in UKB was conducted, nearly 20 principal components (PCs) were included as covariates in the GWAS [2]. PC analysis simplifies the complexity of high-dimensional data while retaining trends and patterns. By transforming the data into fewer dimensions, the data can act as summaries of features (in the given case, kidney function traits). We believe that this approach contributes to the additional reliability of individuals' random allocation regarding comorbidities that occur after birth.

The implementation of MR analysis is predicated on the assumption that a genetic variant, which is determined at conception, is randomly allocated to serve as a valid instrumental variable for the exposure of interest. As MR generally assesses the cumulative effect of a risk factor over a long period of the life course without requiring risk factor assessment and with little change in reverse causation confounded by comorbidities, we believe that consideration of systemic diseases would hardly influence our study.

[1] Sekula P, Del Greco M F, Pattaro C, Köttgen A. Mendelian Randomization as an Approach to Assess Causality Using Observational Data. *J Am Soc Nephrol.* 2016 Nov;27(11):3253-3265. doi: 10.1681/ASN.2016010098.

[2] Stanzick KJ et al. Discovery and prioritization of variants and genes for kidney function in >1.2 million individuals. *Nat Commun.* 2021 Jul 16;12(1):4350. doi: 10.1038/s41467-021-24491-0.

Comment R1-3. It would be great if the authors added more explicit clinical implications of their findings in the discussion part. The discussion section should be completed with these explanations and the generalizability of the findings.

Response R1-3. Thank you for your valuable comment. As we agree that the clinical implications of our findings would be a very interesting issue to discuss, we have elaborated on the potential clinical implications of IL-1ra in protecting kidney function in our Discussion section.

In this study, we investigated the causal effects of various ILs on kidney function, as measured by eGFR, using MR analysis. Our results indicated a significant positive association between genetically predicted IL-1ra levels and kidney function. As IL-1 is a highly active pro-inflammatory cytokine that damages tissues, IL-1 blockade in various autoinflammatory syndromes results in a rapid and sustained reduction in disease severity [1]. IL-1 blockade has proven efficient in treating common conditions in which inflammation is pathogenetically related, such as heart failure, hypertension, and type 2 diabetes [2-4]. Previous studies have reported that inflammatory cytokines including IL-1 are inversely associated with kidney function. In addition, experimental modulation of IL-1 resulted in improved complications of chronic kidney disease, including anemia, in animal models [5]. Clinical trials using IL-1 inhibitors have shown the preventive effects

of IL-1 inhibitors on adverse cardiovascular outcomes. For example, subgroup analysis of the CANTOS trial showed a positive effect of IL-1 inhibitors on the prevention of adverse cardiovascular outcomes in a subgroup of patients with mild CKD [6]. However, this study was not sufficiently powered to examine whether IL-1 inhibition was beneficial for kidney outcomes. The findings of the current study suggest that inhibition of the IL-1 receptor may have clinical potential in protecting kidney function in the general population.

In the revised Discussion section, we have stated that an IL-1 inhibitor could be a potential drug target for protecting kidney function.

Discussion;

IL-1 is a key pro-inflammatory cytokine that activates immune cells and promotes secretion of downstream cytokines including IL-6 and TNF- α , further driving amplification of innate immunity and inflammation in various organs and tissues [7-9] Currently, drugs targeting IL-1, including recombinant IL-1 receptor antagonist (anakinra), monoclonal IL-1 β antibodies (canakinumab), and IL-1 traps (rilonacept) are approved for treatment of inflammation-mediated diseases, including arthritis, gout, type 2 diabetes, and heart failure [1]. In kidney, activated IL-1 pathway causes cell stress, tissue damage, and fibrosis, eventually leading to loss of kidney function [5]. Kidney dysfunction may also be contributed by systemic release of IL-1 in lupus or diabetes that promotes leukocyte adhesion and vascular leakage in the glomeruli [10]. Experimental evidence suggests anti-hypertensive, anti-fibrotic effects of IL-1 receptor blockade in kidney diseases [11-12]. Our observations are in line with findings of these experimental studies, elucidating the important role of IL-1 pathway in kidney function impairment. Furthermore, in a clinical trial of IL-1 inhibitors that primarily investigated the effect of an IL-1 β inhibitor for lowering the risk of hypertension and major adverse cardiovascular events, significant benefits for preventing adverse cardiovascular events were observed in a subgroup of patients with mild CKD [6]. The current study's findings provide MR evidence supporting the use of IL-1 blockade as a potential therapeutic target aimed at preserving kidney function or mitigating the adverse events associated with the deterioration of kidney function among individuals without pre-existing kidney function abnormalities.

[1] Dinarello CA, Simon A, van der Meer JW. Treating inflammation by blocking interleukin-1 in a broad spectrum of diseases. *Nat Rev Drug Discov.* 2012 Aug;11(8):633-52. doi: 10.1038/nrd3800.

[2] Rothman, A. M. *et al.* Effects of Interleukin-1 β Inhibition on Blood Pressure, Incident Hypertension, and Residual Inflammatory Risk: A Secondary Analysis of CANTOS. *Hypertension* **75**, 477-482 (2020).

[3] Rothman AM, MacFadyen J, Thuren T, Webb A, Harrison DG, Guzik TJ, Libby P, Glynn RJ, Ridker PM. Effects of Interleukin-1 β Inhibition on Blood Pressure, Incident

- Hypertension, and Residual Inflammatory Risk: A Secondary Analysis of CANTOS. *Hypertension*. 2020 Feb;75(2):477-482. doi: 10.1161/HYPERTENSIONAHA.119.13642.
- [4] Larsen CM, Faulenbach M, Vaag A, Ehses JA, Donath MY, Mandrup-Poulsen T. Sustained effects of interleukin-1 receptor antagonist treatment in type 2 diabetes. *Diabetes Care*. 2009 Sep;32(9):1663-8. doi: 10.2337/dc09-0533.
- [5] Bandach I, Segev Y, Landau D. Experimental modulation of Interleukin 1 shows its key role in chronic kidney disease progression and anemia. *Sci Rep*. 2021 Mar 18;11(1):6288. doi: 10.1038/s41598-021-85778-2.
- [6] Everett BM, Donath MY, Pradhan AD, Thuren T, Pais P, Nicolau JC, Glynn RJ, Libby P, Ridker PM. Anti-Inflammatory Therapy With Canakinumab for the Prevention and Management of Diabetes. *J Am Coll Cardiol*. 2018 May 29;71(21):2392-2401. doi: 10.1016/j.jacc.2018.03.002.
- [7] Dinarello CA. Biologic basis for interleukin-1 in disease. *Blood*. 1996 Mar 15;87(6):2095-147. PMID: 8630372.
- [8] Garlanda C, Dinarello CA, Mantovani A. The interleukin-1 family: back to the future. *Immunity*. 2013 Dec 12;39(6):1003-18. doi: 10.1016/j.immuni.2013.11.010. PMID: 24332029; PMCID: PMC3933951.
- [9] Sims JE, Smith DE. The IL-1 family: regulators of immunity. *Nat Rev Immunol*. 2010 Feb;10(2):89-102. doi: 10.1038/nri2691. Epub 2010 Jan 18. PMID: 20081871.
- [10] Anders HJ. Of Inflammasomes and Alarmins: IL-1 β and IL-1 α in Kidney Disease. *J Am Soc Nephrol*. 2016 Sep;27(9):2564-75. doi: 10.1681/ASN.2016020177. Epub 2016 Aug 11. PMID: 27516236; PMCID: PMC5004665.
- [11] Lemos DR, McMurdo M, Karaca G, Wilflingseder J, Leaf IA, Gupta N, Miyoshi T, Susa K, Johnson BG, Soliman K, Wang G, Morizane R, Bonventre JV, Duffield JS. Interleukin-1 β Activates a MYC-Dependent Metabolic Switch in Kidney Stromal Cells Necessary for Progressive Tubulointerstitial Fibrosis. *J Am Soc Nephrol*. 2018 Jun;29(6):1690-1705. doi: 10.1681/ASN.2017121283. Epub 2018 May 8. PMID: 29739813; PMCID: PMC6054344.
- [12] Ling YH, Krishnan SM, Chan CT, Diep H, Ferens D, Chin-Dusting J, Kemp-Harper BK, Samuel CS, Hewitson TD, Latz E, Mansell A, Sobey CG, Drummond GR. Anakinra reduces blood pressure and renal fibrosis in one kidney/DOCA/salt-induced hypertension. *Pharmacol Res*. 2017 Feb;116:77-86. doi: 10.1016/j.phrs.2016.12.015. Epub 2016 Dec 13. PMID: 27986554.

Reviewer: 2

Remarks to the Author

In this manuscript, the authors conducted a two-sample MR between interleukins and eGFR in Europeans. MR studies on this topic are scarce and the choice of methods is fine. However, more care is needed in the reporting of methods, results, as well as the discussion of results. Please see below for my concerns:

Abstract

Comment R2-1. Please state IL-1ra in full:

Response R2-1. Thank you for the very important comment. We agree that the terminology used in our abstract should have been presented in its entirety to prevent any confusion, and we have made the necessary modifications as below.

“IL-1 receptor antagonist”

In abstract that provide a summary of the study, we believe that by presenting the full terminology “IL-1 receptor antagonist”, the subject and role of the cytokine that inhibits IL-1 receptor were clarified.

Comment R2-2. Suggest to limit the conclusion to IL-1ra, instead of “IL-1 associated pathway” because results with IL-1 alpha were mostly null:

Response R2-2. Thank you for the very important comment. We agree that the phrase “IL-1 associated pathway” should have been included in our manuscript as below.

“IL-1 receptor antagonist-associated pathway”

We believe that the revised phrase more accurately represents the specific biological pathway. By revising the phrase to “IL-1 receptor antagonist-associated pathway”, the involvement of IL-1 α in the pathway was excluded, thus the scope of the study results could be limited to the specific IL-1ra-associated pathway.

Introduction

Comment R2-3. There is a previous MR study that looked at interleukin and renal functions (PMID: 33393675) and needs to be mentioned, as well as discussed in the discussion.

Response R2-3. Thank you for your valuable comment. We agree that a previous MR study that investigated the association between downregulated serum IL-6 and kidney function should be mentioned and reviewed in the Introduction and Discussion section [1].

A previous MR study was performed to identify the association between genetically proxied IL-6 inhibition and kidney function. The genetic instrument for the downregulation of IL-6 signaling associated with CRP was provided by the UK Biobank. The summary GWAS data for kidney function traits, including log(eGFRcr), blood urea nitrogen, and CKD, were obtained from the CKDGen consortium, which is currently the largest publicly available European ancestry meta-analysis database for kidney function traits. The study demonstrated that there was no strong evidence for an association between genetically proxied inhibition of IL-6 inhibition and log(eGFRcr) through MR analysis, including inversed-variance weighed, median-weighed, and MR-Egger. This study provides MR evidence supporting the safe use of tocilizumab, an IL-6 inhibitor, in patients with renal impairment. Notably, the manufacturer's guidelines for the drug suggest that no dose adjustment is necessary for individuals with mild renal impairment but do not provide recommendations for those with moderate to severe renal dysfunction due to insufficient clinical trial data. Given that patients with CKD are excluded from up to 75% of all randomized controlled trials, the findings from the MR study can provide evidence to address this knowledge gap and offer guidelines for the appropriate use of IL inhibitors in CKD patients [2].

In the revised Introduction section, we have stated the previous MR study that explored the association between genetically proxied IL-6 inhibition and kidney function and briefly noted its clinical implications.

Introduction;

“Mendelian randomization (MR) analysis is a method to estimate the causal effect of exposure on complex traits, overcoming the limitations of conventional observational studies by being less likely to be affected by confounding and reverse causation [3]. Furthermore, MR studies can provide valuable evidence to support the appropriate use of drugs in CKD patients. A previous MR study that did not identify a significant association between IL-6 inhibition and kidney function suggested MR evidence to support the safe administration of tocilizumab, an IL-6 inhibitor, in patients with renal impairment [1]. Quantitative trait locus (QTL) studies enabled discovery of genetic variants that affect a quantitative change of a particular trait, including expression of one or more genes (eQTL) or protein (pQTL) [4-5]. Performing MR analysis with QTL instruments confers insight into identifying candidate genes and molecular targets for diseases by demonstrating the causal effect between gene and protein expression on traits [6-8].”

In the revised Discussion section, we further discuss our study's findings and their relevance to the clinical implications of a previous study that examined the effect of IL-6 inhibition on kidney function. Specifically, we focused on the value of MR studies to provide evidence to support the safe use of IL inhibitors in individuals with CKD.

Discussion;

Another valuable insight produced by implementing MR analysis between genetically proxied IL-1ra and kidney function, in addition to the identification of the underlying mechanisms of cytokine-associated inflammatory pathways and kidney function, is that the evidence for safe administration of pharmacological agents in CKD patients could be provided by MR analysis. For example, in the previous MR study that investigated the association between genetically proxied IL-6 inhibition and kidney function, no significant causal association was found between IL-6 inhibition and kidney function. This finding supports that pharmacological IL-6 inhibition is unlikely to have a direct adverse effect on kidney function [1]. Given that patients with CKD are excluded from up to 75% of all randomized-controlled trials [2], the findings from the MR study can provide reliable evidence for the appropriate use of IL inhibitors in CKD patients. According to studies on IL-1 inhibitors, including the CANTOS or IL-1ra Arthritis Study [9-10], no dose adjustment has been suggested for patients with renal impairment. However, it should be noted that this drug has not been studied in patients with moderate or severe kidney dysfunction. Further clinical trials are necessary to investigate the safety of IL-1 inhibitors in patients with impaired kidney function, and the current study suggests the possible safe administration of IL-1 inhibitors in CKD patients by providing MR evidence.

We believe that the discussion of the previous study of IL-6 and addressing the importance of MR studies in providing evidence of drug safety in CKD patients could provide readers with a more comprehensive understanding of the current study and its implications for clinical practice.

[1] Ryan DK, Karhunen V, Walker DJ, Gill D. Inhibition of interleukin 6 signalling and renal function: A Mendelian randomization study. *Br J Clin Pharmacol*. 2021 Jul;87(7):3000-3013. doi: 10.1111/bcp.14725. Epub 2021 Feb 10. PMID: 33393675; PMCID: PMC8327328.

[2] Charytan D, Kuntz RE. The exclusion of patients with chronic kidney disease from clinical trials in coronary artery disease. *Kidney Int*. 2006 Dec;70(11):2021-30. doi: 10.1038/sj.ki.5001934. Epub 2006 Oct 18. PMID: 17051142; PMCID: PMC2950017.

[3] Smith GD, Ebrahim S. 'Mendelian randomization': can genetic epidemiology contribute to understanding environmental determinants of disease? *Int J Epidemiol*. 2003 Feb;32(1):1-22. doi: 10.1093/ije/dyg070.

[4] Yao DW, O'Connor LJ, Price AL, Gusev A. Quantifying genetic effects on disease mediated by assayed gene expression levels. *Nat Genet*. 2020 Jun;52(6):626-633. doi: 10.1038/s41588-020-0625-2. Epub 2020 May 18.

[5] Vösa U et al., Large-scale cis- and trans-eQTL analyses identify thousands of genetic loci and polygenic scores that regulate blood gene expression. *Nat Genet.* 2021 Sep;53(9):1300-1310. doi: 10.1038/s41588-021-00913-z. Epub 2021 Sep 2.

[6] Lawrenson K et al., Cis-eQTL analysis and functional validation of candidate susceptibility genes for high-grade serous ovarian cancer. *Nat Commun.* 2015 Sep 22;6:8234. doi: 10.1038/ncomms9234. PMID: 26391404

[7] Zhu Z et al., Integration of summary data from GWAS and eQTL studies predicts complex trait gene targets. *Nat Genet.* 2016 May;48(5):481-7. doi: 10.1038/ng.3538.

[8] Karhunen V, G. D., Malik R, Ponsford MJ, Ahola-Olli A, Papadopoulou A, et al. Genetic study of circulating cytokines offers insight into the determinants, cascades and effects of systemic inflammation. *medRxiv* (2020).

[9] Ridker PM et al., CANTOS Trial Group. Antiinflammatory Therapy with Canakinumab for Atherosclerotic Disease. *N Engl J Med.* 2017 Sep 21;377(12):1119-1131. doi: 10.1056/NEJMoa1707914. Epub 2017 Aug 27. PMID: 28845751.

[10] Campion GV, Lebsack ME, Lookabaugh J, Gordon G, Catalano M. Dose-range and dose-frequency study of recombinant human interleukin-1 receptor antagonist in patients with rheumatoid arthritis. The IL-1Ra Arthritis Study Group. *Arthritis Rheum.* 1996 Jul;39(7):1092-101. doi: 10.1002/art.1780390704. PMID: 8670316.

Comment R2-4. MR can't minimise confounding and reverse causation, but effect estimates obtained using MR are less likely to be affected by confounding and reverse causation. Please amend

Response R2-4. We truly appreciate your comment. We agree that MR cannot minimize confounding and reverse causation and should have been included in our manuscript as below.

“Mendelian randomization (MR) analysis is a method to estimate the causal effect of exposure on complex traits, overcoming the limitations of conventional observational studies by being less likely to be affected by confounding and reverse causation [1].”

We believe that the revised phrase accurately describes the MR methodology's advantages, which have little chance to be affected by confounding and reverse causation.

[1] Smith, G. D. & Ebrahim, S. 'Mendelian randomization': can genetic epidemiology contribute to understanding environmental determinants of disease? *Int J Epidemiol* **32**, 1-22 (2003).

Method

Comment R2-5. Since there are a lot of ILs, please explain how the authors arrived at the final list of ILs

Response R2-5. Thank you for your comments. In the kidney, an important role in homeostasis and disease progression is attributed to ILs, which perform several biological functions and interact with other cells and tissues [1]. Please note that We used a previously developed cis-instrument for ILs, which are genetic variants located in or close to the coding gene that is naturally more relevant to the expression of that gene in comparison to other genes, to perform MR analysis between serum IL levels and kidney function [2]. Among various ILs developed from the previous study, the cis-QTL instruments of IL-1 α , IL-1ra, IL-2ra, IL-6, IL-7, IL-8, IL-12p70, IL-16, and IL-18 were MR analyzed to examine their association with kidney function.

Please note that Each IL is pathogenetically associated with kidney inflammation, fibrosis, and decline in kidney function.

1. IL-1 α and IL-6 are associated with kidney tissue fibrosis and inversely associated with measures of kidney function. IL-1 α is activated by inflammasome which contributes to kidney inflammation and systemic response in immune cells [3-5].
2. IL-1ra is a key inflammatory modulator, as IL-1ra knockout mice develop a spontaneous chronic inflammatory status with eGFR decline [6].
3. IL-2 is involved in the homeostasis of regulatory T-cells and IL-2 deficiency has been reported to be associated with inflammatory kidney diseases such as lupus nephritis [7].
4. IL-7 is related to renal proximal tubule epithelial cell fibrosis, and IL-8, also known as CXCL8, is a deleterious chemokine involved in renal injury in glomerular diseases [8].
5. IL-12 is produced by macrophages and dendritic cells, and in conjugation with IL-18, it promotes interferon- γ production and proliferation of naïve T-cells. Serum IL-12 and IL-18 are associated with glomerulonephritis and proteinuria [9].
6. Lastly, IL-16 is a well-known cytokine associated with ischemia-reperfusion injury [10].

We believe that the ILs analyzed in this study should be investigated using MR analysis of kidney function traits, considering the potential impacts of these ILs on kidney function. In the revised Methods section, we have stated that the analyzed ILs were associated with the pathogenesis of kidney inflammation, fibrosis, and injury.

In our Methods;

“The examined ILs were previously reported to be associated with the pathogenesis of kidney inflammation, fibrosis, and injury [1-10].”

Please understand that stating the association between each IL other than IL-1ra and kidney function in the Discussion section may interrupt the emphasis on the protective effect of IL-1ra on kidney function and its clinical implications. Thus, we have maintained our primary focus on IL-1ra in the Discussion section. If you consider that an additional description would be needed, please let us know, and we will further provide the previously reported effect of various ILs on kidney function.

- [1] Mertowska, P., Mertowski, S., Smarz-Widelska, I. & Grywalska, E. Biological Role, Mechanism of Action and the Importance of Interleukins in Kidney Diseases. *Int J Mol Sci* **23** (2022)
- [2] Bouras, E. *et al.* Circulating inflammatory cytokines and risk of five cancers: a Mendelian randomization analysis. *BMC Med* **20**, 3 (2022).
- [3] Anders, H. J. Of Inflammasomes and Alarmins: IL-1 β and IL-1 α in Kidney Disease. *J Am Soc Nephrol* **27**, 2564-2575 (2016).
- [4] Netea, M. G., van de Veerdonk, F. L., van der Meer, J. W., Dinarello, C. A. & Joosten, L. A. Inflammasome-independent regulation of IL-1-family cytokines. *Annu Rev Immunol* **33**, 49-77 (2015).
- [5] Yazdi, A. S. & Drexler, S. K. Regulation of interleukin 1 α secretion by inflammasomes. *Ann Rheum Dis* **72 Suppl 2**, ii96-99 (2013).
- [6] Bandach, I., Segev, Y. & Landau, D. Experimental modulation of Interleukin 1 shows its key role in chronic kidney disease progression and anemia. *Sci Rep* **11**, 6288 (2021).
- [7] Rose, A. *et al.* IL-2 Therapy Diminishes Renal Inflammation and the Activity of Kidney-Infiltrating CD4+ T Cells in Murine Lupus Nephritis. *Cells* **8** (2019). <https://doi.org:10.3390/cells8101234>
- [8] Hsieh, P. F. *et al.* The role of IL-7 in renal proximal tubule epithelial cells fibrosis. *Mol Immunol* **50**, 74-82 (2012). <https://doi.org:10.1016/j.molimm.2011.12.004>
- [9] Tucci, M., Lombardi, L., Richards, H. B., Dammacco, F. & Silvestris, F. Overexpression of interleukin-12 and T helper 1 predominance in lupus nephritis. *Clin Exp Immunol* **154**, 247-254 (2008). <https://doi.org:10.1111/j.1365-2249.2008.03758.x>
- [10] Wang, S. *et al.* Decreased renal ischemia-reperfusion injury by IL-16 inactivation. *Kidney Int* **73**, 318-326 (2008). <https://doi.org:10.1038/sj.ki.5002692>

Comment R2-6. Suggest not to mention the component studies of meta-analysis to reduce confusion

Response R2-6. Thank you for your comments. We agree that the information on the component studies should have been excluded from our manuscript as follows.

In our Methods;

“Genetic instrument for MR analysis

Genetic instruments were developed from a previous GWAS meta-analysis study including genetic information of 47 inflammatory cytokines [1]. Using these genetic instruments, a total of 80 cis-instruments (60 as cis-pQTL and 20 as cis-eQTL) that are strongly associated with the effect of circulating IL concentration were developed (Supplemental Table 1,2) [1]. Two cis-instrumental selection criteria were used to identify variants that reflected the effect of circulating cytokine levels and are described in Supplemental Data 1.”

We believe that the potential confusion that might have been caused by listing every meta-analysis study in the development of genetic instruments for ILs could be avoided.

[1] Bouras, E. *et al.* Circulating inflammatory cytokines and risk of five cancers: a Mendelian randomization analysis. *BMC Med* **20**, 3 (2022).

Comment R2-7. Please explain if clumping was involved when selecting genetic instruments and justify if not

Response R2-7. Thank you for your comments. In agreement with your comment, we have added the involvement of clumping when selecting the genetic instruments in the revised manuscript. It should be noted that clumping was used in a previous study that identified the cis-instrument of ILs [1].

Directly cited

“In the context of a cis-region MR, using a very small correlation threshold may result in a loss of causal variants; therefore, clumping was performed using a pairwise linkage disequilibrium (LD) threshold of $r^2 < 0.1$.”

In our Methods;

“To mitigate the potential loss of causal variants in a cis-region MR study due to a small correlation threshold, clumping was performed with a pairwise linkage disequilibrium threshold of $r^2 < 0.1$.”

[1] Bouras, E. *et al.* Circulating inflammatory cytokines and risk of five cancers: a Mendelian randomization analysis. *BMC Med* **20**, 3 (2022).

Comment R2-8. Please provide a reference for using the median of r^2 to depict strength of instruments. This method could possibly include weak instruments, particularly some instruments in the supplemental tables had F-stat below 10, implying results could be biased by weak instruments.

Response R2-8. Thank you for your comments. We agree that this study included weak instruments, as some instruments had an F-statistic < 10 . In this study, we evaluated the strengths of the genetic instruments using both the F-statistic and the proportion of variance explained (r^2) for each genetic variant. It should be noted that both F and r^2 statistics from the regression of the risk factor on the instrumental variants (IVs) based on a large population can be used to judge the strength of the instrumental variants [1-2]. The use of r^2 statistics to address instrumental power is well described in the cited literature [2].

Directly cited

*“For MR studies using multiple variants, we have described the relationships among numerous key study variables, including the number of variants, variant effect sizes, exposure effect sizes, R^2 , adjusted R^2 , F, and power. Our results suggest that the **power to detect a causal effect depends strongly on the R^2 value** of the first-stage regression (not the adjusted R^2) and is not influenced by allele frequencies or the number of IVs included in a regression.”*

Please understand that we are very aware of some of our weak-powered instruments, with F-statistics below 10. As such, we mentioned that instrumental power was not secured in the analysis, and false-negative bias may be present in the limitations section. Specifically, regarding your comment, we have added that despite the evaluation of F-statistics and r^2 for the assessment of the strengths of instruments, there is a potential for false-negative bias due to weak-powered instruments with F-statistics < 10 . This is connected to the following request: one must not overlook the possibility of the potential effect of serum ILs on kidney function based on the null findings reported in the study. This is stated in the limitations section with the appropriate reference as below.

In Discussion;

“Our study has some limitations. First, the number of genetic instruments was insufficient for some ILs, which restricted the type of sensitivity analysis that could be performed. Although the strengths of the instruments were evaluated using F-statistics and r^2 [2], there were weak-powered instruments with F-statistics <10. It should be noted that there is potential for false-negative bias, particularly when the true effect is below the minimum effect size (Supplemental Table 6). Thus, one may not preclude the possibility of a kidney function effect based on the null findings in this study.”

In addition, we provided the minimum effect size of each IL for a power of 80% using the meta-analysis dataset for log(eGFRcr) of the CKDGen and UKB datasets, and r^2 values at a type-I error rate of 0.25, as shown in Supplemental Table 6. We provided a minimum effect size to suggest the potential for false-negative bias in instrumental variables with a true effect below the minimum effect size. Please refer to Response R3-3 for further explanation of the minimum effect size.

[1] Palmer, T. M. et al. Using multiple genetic variants as instrumental variables for modifiable risk factors. *Stat Methods Med Res* **21**, 223-242 (2012).
<https://doi.org/10.1177/0962280210394459>

[2] Pierce BL, Ahsan H, Vanderweele TJ. Power and instrument strength requirements for Mendelian randomization studies using multiple genetic variants. *Int J Epidemiol.* 2011 Jun;40(3):740-52. doi: 10.1093/ije/dyq151. Epub 2010 Sep 2. PMID: 20813862; PMCID: PMC3147064.

Comment R2-9. Please clarify the source of exposure data. It was mentioned that the analysis was independently replicated using CKDGen and UKBiobank, please state which outcome was replicated as this was not shown in Figure 1.

Response R2-9. Thank you for your comments. Please note that the exposure data used in this study were obtained from a previous study that investigated the association between circulating inflammatory cytokines and the risk of five cancers [1]. We agree that clarification of the source of exposure data is needed; thus, we demonstrated the source of exposure and outcome data in both our Methods sections in a more precise way to avoid confusion. The genetic instruments were identified from three GWAS sources: (1) GWAS of the SCALLOP consortium (N = 21,758), INTERVAL study (N = 3,301), and Finnish consort (N = 13,365) [2-6]. Please understand that the addition of all exposure data in figure 1 may increase the complexity and hinder a focused and concise description of the study setting.

Additionally, in agreement with your comment, we have indicated the datasets that were analyzed for the replication purposes of the main data in figure 1. The replication datasets were (1) CKDGen meta-analysis of log(eGFRcr) and (2) CKDGen and UKB meta-analysis of log(eGFRcys). Below the boxes of each dataset, we indicate either “*Main analysis*” or

“Replication”. Please note that the analysis conducted using summary data of annual eGFR decline was not a replication of the MR analysis for log(eGFRcr), but an individual MR analysis to identify the association between genetically predicted serum levels of each IL and annual eGFR decline. Therefore, the annual eGFR decline dataset is not indicated in the replication analysis in figure 1.

Summary of the revision

In our Methods;

“Study setting

We performed a two-sample summary-level MR analysis of the serum IL levels and kidney function traits. The genetic instruments for serum IL levels were developed from the previous study [1]. The outcome GWAS summary statistics for kidney function traits were obtained from the CKDGen consortium, which is widely used for MR analysis towards kidney function traits (Figure 1)[2-4].”

In Figure 1;

Figure legend;

“Below the boxes of each dataset, it is indicated whether the datasets were analyzed as “Main analysis” or “Replication”.”

- [1] Bouras, E. *et al.* Circulating inflammatory cytokines and risk of five cancers: a Mendelian randomization analysis. *BMC Med* **20**, 3 (2022).
- [2] Folkersen, L. *et al.* Genomic and drug target evaluation of 90 cardiovascular proteins in 30,931 individuals. *Nat Metab* **2**, 1135-1148 (2020). [https://doi.org:10.1038/s42255-020-00287-2](https://doi.org/10.1038/s42255-020-00287-2)
- [3] Sun, B. B. *et al.* Genomic atlas of the human plasma proteome. *Nature* **558**, 73-79 (2018). [https://doi.org:10.1038/s41586-018-0175-2](https://doi.org/10.1038/s41586-018-0175-2)
- [4] University of Oulu. Northern Finland Birth Cohort 1966.
- [5] Sliz, E. *et al.* Genome-wide association study identifies seven novel loci associating with circulating cytokines and cell adhesion molecules in Finns. *J Med Genet* **56**, 607-616 (2019). [https://doi.org:10.1136/jmedgenet-2018-105965](https://doi.org/10.1136/jmedgenet-2018-105965)
- [6] Karhunen V, G. D., Malik R, Ponsford MJ, Ahola-Olli A, Papadopoulou A, et al. Genetic study of circulating cytokines offers insight into the determinants, cascades and effects of systemic inflammation. *medRxiv* (2020)
- [7] Park, S. *et al.* Short or Long Sleep Duration and CKD: A Mendelian Randomization Study. *J Am Soc Nephrol* **31**, 2937-2947 (2020). [https://doi.org:10.1681/asn.2020050666](https://doi.org/10.1681/asn.2020050666)
- [8] Park, S. *et al.* A Mendelian randomization study found causal linkage between telomere attrition and chronic kidney disease. *Kidney Int* **100**, 1063-1070 (2021). [https://doi.org:10.1016/j.kint.2021.06.041](https://doi.org/10.1016/j.kint.2021.06.041)
- [9] Park, S. *et al.* Atrial fibrillation and kidney function: a bidirectional Mendelian randomization study. *Eur Heart J* **42**, 2816-2823 (2021). [https://doi.org:10.1093/eurheartj/ehab291](https://doi.org/10.1093/eurheartj/ehab291)

Comment R2-10. For creatinine-based log eGFR, I wonder what is the value added of using CKDGen alone alongside of the meta-analysis of CKDGEN + UKB, which is supposed to be of greater statistical power?

Response R2-10. Thank you for your comments. In agreement with your observation, we conducted the main analysis using the largest GWAS dataset available for log(eGFRcr), which is a meta-analysis of CKDGen and UKB, owing to the greater statistical power with a low possibility of weak instrumental bias. However, please note that the UKB dataset has a healthy volunteer bias owing to the lower CKD prevalence. The UKB dataset may reflect a higher proportion of individuals with normal eGFR values [1]. Therefore, there was a need to separate

the CKDGen database, conduct additional analyses, and conduct a replication analysis using CKDGen data, which we believe augments the value of our study. Nevertheless, we still believe that presenting MR results from the largest and most recent dataset as the main analysis is appropriate, as large population-based outcome data provide greater statistical power and a low possibility of weak instrumental bias. Thus, we presented the MR results from a meta-analysis of log(eGFRcr) from the CKDGen + UKB datasets as our main analysis. To provide additional support for our findings, we included two replication analyses using the following datasets: log(eGFRcr) from CKDGen, and log(eGFRcys) from CKDGen + UKB. In the revised Methods section, we stated that the main analysis was replicated with two different GWAS meta-analysis datasets for the possibility of healthy volunteer bias due to the lower CKD prevalence in the UKB dataset. As MR relies on some untestable assumptions, consistent findings in two independent datasets and relevant outcomes (eGFR-Cr, eGFR-cystatin C, and eGFR change) support the validity of the findings and are less likely to be biased from false-positive findings.

In Methods;

“Kidney function outcome in the MR analysis

We used three eGFR outcome datasets as eGFR values are a current standard parameter for the assessment of kidney function; 1) the meta-analysis for creatinine-based log-eGFR values of the CKDGen and UKB data from 2021 (n = 765,348) which has strength as this is the currently largest GWAS meta-analysis for eGFR trait [2], 2) creatinine-based log-eGFR values from the phase 4 CKDGen study from 2019 (n = 435,581) [3], and 3) cystatin C-based log-eGFR values from the CKDGen and UKB (n = 460,826) from 2021 [2], which has particular strength in that the dataset can be used to test the causal estimates towards cystatin C-based eGFR values which is less affected from non-kidney factors than creatinine-based levels. The main analysis was performed using the largest GWAS meta-analysis of creatinine-based log-eGFR from CKDGen and UKB because of the greater statistical power and lower possibility of weak-instrumental bias. Nevertheless, considering the potential for healthy volunteer bias resulting from the relatively lower prevalence of CKD in the UKB dataset [1], replication analyses employing the other two GWAS meta-analysis datasets were conducted. Additionally, the UKB dataset was independent of the CKDGen data, fulfilling an independent replication analysis, and was independent of the samples included in the GWAS for genetic instrument development. Such a two-sample MR without participant overlap has statistical power even if the results are affected by instrumental power because the potential bias would be towards false-negative findings. Therefore, a positive finding from such an independent two-sample MR is more likely to reflect true causal effects [4].”

[1] Bycroft, C. et al. The UK Biobank resource with deep phenotyping and genomic data. *Nature* **562**, 203-209 (2018). [https://doi.org:10.1038/s41586-018-0579-z](https://doi.org/10.1038/s41586-018-0579-z)

[2] Stanzick, K. J. et al. Discovery and prioritization of variants and genes for kidney function in >1.2 million individuals. *Nat Commun* **12**, 4350 (2021).

<https://doi.org/10.1038/s41467-021-24491-0>

[3] Wuttke, M. *et al.* A catalog of genetic loci associated with kidney function from analyses of a million individuals. *Nat Genet* **51**, 957-972 (2019). <https://doi.org/10.1038/s41588-019-0407-x>

[4] Burgess, S., Davies, N. M. & Thompson, S. G. Bias due to participant overlap in two-sample Mendelian randomization. *Genet Epidemiol* **40**, 597-608 (2016). <https://doi.org/10.1002/gepi.21998>

Comment R2-11. For cystatin C-based log eGFR, it is stated in the method that the data was from UKB, but in Figure 1 it is shown as CKDGen + UKB. Please clarify the data source.

Response R2-11. Thank you for your very important comment and we agree that the information should have been included in our manuscript as below. Please note that meta-analysis of cystatin-C based log(eGFR) was performed using CKDGen and UKB data, and the number of individuals included in the analysis was 460,826 [1].

Directly cited

“For the alternative biomarker support, we conducted analyses in UKB and meta-analysed these results with CKDGen association results for eGFRcys and BUN (n = 460,826 and 852,678, respectively).”

In the revised manuscript, we clarified that the meta-analysis was based on CKDGen and UKB data and the included number of participants. In addition, we revised the figure and figure legend. For the figure, please refer to **Response R2-9**.

In Methods;

“3) cystatin C-based log-eGFR values from the CKDGen and UKB (n = 460,826)”

In Figure legend;

“the meta-analysis for creatinine-based log-eGFR values of CKDGen and UKB (n = 765,348), creatinine-based log-eGFR values from the phase 4 CKDGen study (n = 436,581), cystatin C-based log-eGFR values from CKDGen and UKB (n = 460,826), and degree of annual eGFR decline from CKDGen and UKB (n = 343,339).”

[1] Stanzick, K. J. *et al.* Discovery and prioritization of variants and genes for kidney function in >1.2 million individuals. *Nat Commun* **12**, 4350 (2021).
<https://doi.org/10.1038/s41467-021-24491-0>

Comment R2-12. Please provide reference for the sentence “cystatin C-based eGFR values which is less affected from non-kidney factors than creatinine-based levels”.

Response R2-12. Thank you for your comments. Please note that The identification of genetic factors that contribute to kidney function is a challenging task, and one of the difficulties is the dissection of the eGFRcr loci likely related to kidney function from those related to creatinine metabolism [1]. As creatinine is released from the breakdown of muscle tissue, serum creatinine levels are highly influenced by body muscle mass, dietary habits, and body habitus. On the other hand, cystatin C is a small protein produced by all nucleated cells and is freely filtered at the glomerulus; therefore, it is less susceptible to individual characteristics of the patient and is not influenced by muscle mass, age, gender, or ethnicity. In addition, kidney function assessment using eGFRcys is superior to eGFRcr in measuring kidney function and predicting morbidity and mortality [2]. Genetic data on eGFRcys were used to assess the consistency of the effects [1].

Nevertheless, please note that the preference for eGFRcr over eGFRcys in clinical practice can be attributed to economic considerations. Specifically, the cost of the reagents necessary for cystatin C testing is approximately ten times higher than that for serum creatinine testing. Consequently, eGFRcr is often chosen as a more cost-effective option for kidney function assessment, despite its limitations in accuracy and sensitivity compared to eGFRcys [2].

Following your suggestion, we have added the reference for the statement that cystatin C-based eGFR values are less affected by non-kidney factors than creatinine-based levels in the Methods section as below.

In our Methods;

“cystatin C-based eGFR values which is less affected from non-kidney factors than creatinine-based levels [2].”

[1] Stanzick, K. J. *et al.* Discovery and prioritization of variants and genes for kidney function in >1.2 million individuals. *Nat Commun* **12**, 4350 (2021).
<https://doi.org/10.1038/s41467-021-24491-0>

[2] Lees, J. S. *et al.* Glomerular filtration rate by differing measures, albuminuria and prediction of cardiovascular disease, mortality and end-stage kidney disease. *Nat Med*

25, 1753-1760 (2019). <https://doi.org/10.1038/s41591-019-0627-8>

Comment R2-13. Please state clearly that GWAS of annual eGFR decline is a meta-analysis of data from CKDGen and UKB.

Response R2-13. Thank you for the comment. We agree that the GWAS of annual eGFR decline is a meta-analysis of data from CKDGen and UKB should be stated more clearly. The statement of the revised manuscript is as below.

In our Methods;

“Furthermore, to test whether IL levels have causal effects on accelerated eGFR decline rather than on a static value, we implemented MR analysis using GWAS meta-analysis of the degree of annual eGFR decline from CKDGen and UKB (n = 343,339) as another outcome dataset [1].”

[1] Gorski, M. *et al.* Genetic loci and prioritization of genes for kidney function decline derived from a meta-analysis of 62 longitudinal genome-wide association studies. *Kidney Int* **102**, 624-639 (2022). <https://doi.org/10.1016/j.kint.2022.05.021>

Comment R2-14. Please explain specifically what was done to “ascertain the direction of the variants’ effects” (P.9), as this sentence does not seem to be relevant to the next sentence.

Response R2-14. Thank you for your comments. We agree that the specific method used to ascertain the direction of variant effects should have been stated after the sentence. Please note that some scenarios of genetic association with the causal trait can lead to existing causal inference methods giving the wrong direction of causality; therefore, a method such as Steiger filtering is needed to ascertain the direction of the variants’ effects [1]. Steiger filtering can be applied to summary-level data and is potentially less susceptible to problems, such as measurement errors. In agreement with your suggestion, we have stated the method to determine the direction of the variant effects in the Methods section of the revised manuscript. In addition, we present the Steiger filtering results for each instrumental variable in Supplemental Table 2.

In our Methods;

“MR assumptions

The directions of the variants' causal effects were ascertained using MR Steiger filtering [1] to establish that there is no direct effect of the genetic instruments on kidney function traits, which may potentially violate the exclusion-restriction assumption. Steiger filtering results for each instrumental variant are presented in Supplemental Table 2, and all variants indicate evidence of causality in the expected direction.”

[1] Hemani G, Tilling K, Davey Smith G. Orienting the causal relationship between imprecisely measured traits using GWAS summary data. PLoS Genet. 2017 Nov 17;13(11):e1007081. doi: 10.1371/journal.pgen.1007081. Erratum in: PLoS Genet. 2017 Dec 29;13(12):e1007149. PMID: 29149188; PMCID: PMC5711033.

Comment R2-15. MR Egger could not provide a “consistent” estimate. Do the authors mean “valid”?

Response R2-15. Thank you for the comment. We agree that the MR-Egger could not provide “consistent” but “valid” estimate and we have made the necessary modifications as below.

In our Methods;

*“First, MR-Egger regression analysis with bootstrapped standard error provides a pleiotropy-robust causal estimate that is **valid** even if all genetic variants are invalid due to violation of the independence assumption [1].”*

We believe that the revised word has represented in a more precise explanation for the meaning of MR-Egger analysis.

[1] Burgess, S. *et al.* Guidelines for performing Mendelian randomization investigations. *Wellcome Open Res* **4**, 186 (2019). <https://doi.org/10.12688/wellcomeopenres.15555.2>

Comment R2-16. The ability to calculate P of intercept in MR Egger is not a strength. Please delete

Response R2-16. Thank you for the comment. We agree that the ability to calculate P of intercept in MR Egger is not a strength. The revised manuscript excluded the description by strength.

In our Methods;

“A significant MR-Egger intercept P value would raise the suspicion of directional pleiotropy.”

Comment R2-17. Please explain how the exclusion of cis-eQTL instruments would improve the analysis

Response R2-17. Thank you for your valuable comment. Please note that the expression quantitative trait loci (eQTL) analysis attempts to identify the genomic locations that influence variation in gene expression levels (mRNA abundance) [1]. An eQTL near the genomic location of the influenced gene is called cis-eQTL, and an eQTL far away from the influenced gene is called trans-eQTL. It is important to note that because an eQTL can influence the expression of multiple genes, pleiotropy due to an eQTL may occur [1-2]. This indicates that the use of eQTL as genetic variants may affect the outcome trait through pathways other than the exposure of interest, leading to a violation of the exclusion-restriction assumption.

In this context, the exclusion of cis-eQTL instruments in MR analysis can serve as a sensitivity analysis to assess the validity of the causal inference made in MR analysis. We believe that this approach enables us to assess the potential bias due to potential pleiotropy produced from eQTL instruments and validate the causal inference derived from MR analysis. In the revised Methods section, we have stated that the exclusion of cis-eQTL instruments can be used for sensitivity analysis.

Methods;

“Other statistical aspects of the MR analysis

We performed two additional sensitivity MR analyses: 1) excluding cis-eQTL instruments and 2) excluding palindromic SNPs to test whether differences in instrument development methods or uncertain harmonization might have influenced the results. As cis-eQTL instruments are near the genomic location of the influenced genes that can influence the expression of multiple genes, potential pleiotropy due to cis-eQTL instruments can occur [1-2]. Therefore, sensitivity analysis with the exclusion of cis-eQTL instruments was implemented to assess the validity of the causal inference.”

[1] Tian, J. *et al.* The Dissection of Expression Quantitative Trait Locus Hotspots. *Genetics* **202**, 1563-1574 (2016). <https://doi.org/10.1534/genetics.115.183624>

[2] Liu, Y. *et al.* Genome-wide analysis of expression QTL (eQTL) and allele-specific expression (ASE) in pig muscle identifies candidate genes for meat quality traits. *Genet Sel Evol* **52**, 59 (2020). <https://doi.org/10.1186/s12711-020-00579-x>

Comment R2-18. Why weren't palindromic SNPs excluded in the main analysis, as done in default in the "TwoSampleMR" package?

Response R2-18. Thank you for your comments. We agree that palindromic SNPs can be excluded from the main analysis as it is the default in the "TwoSampleMR" package. We are very aware that additional care should be taken for palindromic variants to verify whether the allele originated from a particular allele. This is because not all publicly available data resources are consistent in correctly reporting strand information. However, please note that there is no need to assume that strandedness differs among recent GWAS data. Modern GWAS typically correct for mis-strandedness prior to facilitating a meta-analysis [1].

Please note that there are some options to deal with palindromic SNPs, including 1) replacing them with non-palindromic linkage disequilibrium proxies, 2) conducting sensitivity analyses to evaluate the impact of palindromic SNPs on MR results, or 3) excluding them [2]. Because the exclusion of SNPs can be considered for data manipulation, we were careful in excluding palindromic SNPs. Thus, we conducted an analysis under two scenarios, including or excluding palindromic SNPs, to evaluate the robustness of our results.

Please understand that Despite the fact that we are aware of the limitations of weak instruments, we aimed to conserve SNPs whenever possible so as not to exclude significant signals in the MR analysis.

[1] Zhang L et al., Multistage genome-wide association meta-analyses identified two new loci for bone mineral density. *Hum Mol Genet.* 2014 Apr 1;23(7):1923-33. doi: 10.1093/hmg/ddt575. Epub 2013 Nov 17. PMID: 24249740; PMCID: PMC3943521.

[2] Hartwig, F. P., Davies, N. M., Hemani, G. & Davey Smith, G. Two-sample Mendelian randomization: avoiding the downsides of a powerful, widely applicable but potentially fallible technique. *Int J Epidemiol* **45**, 1717-1726 (2016). <https://doi.org/10.1093/ije/dyx028>

Comment R2-19. The authors stated that they considered the issue of multiple comparison but used a p-value threshold of 0.05 for statistical significance. Please further elaborate on this.

Response R2-19. Thank you for your comments. We agree that there is an increased possibility of false-positive results, also known as type I errors, as multiple types of ILs were used for MR analysis. In addition, we agree that a P value threshold of 0.05 could be regarded as inappropriate for statistical significance. However, please note that we performed various sensitivity analyses and only considered an MR result significant when the significance threshold of $P < 0.05$ across all the sensitivity analyses was met. This point has been clearly included in the Methods section.

In Methods;

“Other statistical aspects of the MR analysis

We considered the possibility of false-positive findings from multiple comparisons as the causal estimates were tested for multiple types of ILs. To address this issue, we performed multiple sensitivity analyses and only considered an MR result as a true positive signal when the significance threshold was $P < 0.05$, across all the sensitivity analyses performed per IL.”

Result

Comment R2-20. It was stated in the method that the eGFR values were logged in the outcome GWAS, but in the results the authors reported the change in eGFR as percentages. Please explain.

Response R2-20. Thank you for your valuable comment. We agree that the unnoticed change in reporting the eGFR change from logged value to percentage may cause confusion for readers. Please understand that We converted the change in the log(eGFR) value into a percentage change with the intention of presenting results that are more easily understandable and acceptable for medical practitioners.

We agree that the explanation for representing the result should be made; thus, we have amended the Methods section of our manuscript. Specifically, we have included an explanation for the use of the change in percentage (with standard error) to describe the change in log(eGFR) by the genetically predicted standard deviation in serum IL concentration. We believe that the revised statement ensured that the reason for the change in reporting formats can be clearly and easily understood by readers.

In Methods;

“Two-sample MR analysis with summary-level data

*Summary-level MR analysis was performed using two sets of genetic instruments (cis-pQTLs and cis-eQTLs) for each IL on the four outcome datasets. First, we implemented a multiplicative random-effect inverse variance weighted (IVW) method when more than one genetic instrument was available to construct an instrumental variable for a given IL. Provided with only a single genetic instrument, the ratio of coefficient, also referred to as the Wald ratio, was applied to measure MR estimates [1]. Multiplicative random-effects model is the current standard for main MR analysis as the method allows balanced pleiotropic effects [2-3]. In addition, leave-one-out analysis, leaving one single nucleotide polymorphism (SNP) out of the analysis at a time, were performed to inspect whether there was bias from an extreme outlier suspected with pleiotropic effect. **The MR estimates were converted into the degree of change (percentage [standard error]) of log-transformed eGFR according to the genetically predicted standard deviation in serum IL concentration to facilitate interpretation.**”*

[1] Burgess, S., Butterworth, A. & Thompson, S. G. Mendelian randomization analysis with multiple genetic variants using summarized data. *Genet Epidemiol* **37**, 658-665 (2013).

[2] Burgess, S. *et al.* Guidelines for performing Mendelian randomization investigations. *Wellcome Open Res* **4**, 186 (2019).

[3] Bowden, J. *et al.* Improving the accuracy of two-sample summary-data Mendelian randomization: moving beyond the NOME assumption. *Int J Epidemiol* **48**, 728-742 (2019).

Comment R2-21. Please state where the results of cystatin-C-based eGFR, as well as those of MR-Egger, were presented (which supp table?)

Response R2-21. Thank you for your comments. Please note that cystatin-C-based log(eGFR) and its MR-Egger are reported in each table's "Outcome" column, and we have presented the source of the outcome summary data in the table's legend. Specifically, the MR estimates of cystatin C-based log(eGFR) are presented below the row of the creatinine-based log(eGFR).

However, as multiple outcome GWAS data were used for the analysis, further clarification is needed. In the revised manuscript, we have presented the source of the outcome data in the table and supplemental table, as well as each table's legends.

In Tables;

Interleukin	*Outcome	MR-Egger intercept P	MR methods	eGFR change beta (%)	Standard error (%)	P value
IL-1ra	1) Creatinine-based log-eGFR values (CKDGen + UKB)	0.16	MR-IVW	0.28	0.11	0.009
			MR-Egger	0.36	0.12	0.003
			Weighed median	0.32	0.11	0.004
	2) Creatinine-based log-eGFR values (CKDGen)	0.53	MR-IVW	0.32	0.09	3E-04
			MR-Egger	0.25	0.14	0.04
			Weighed median	0.36	0.13	0.005
	3) Cystatin-C-based log-eGFR values (CKDGen + UKB)	0.16	MR-IVW	0.35	0.17	0.039
			MR-Egger	0.73	0.21	0.001
			Weighed median	0.52	0.18	0.003
	4) Degree of annual eGFR decline (CKDGen + UKB)	0.29	MR-IVW	-2.18	1.09	0.043
			MR-Egger	-2.42	1.71	0.078
			Weighed median	-2.85	1.48	0.049

^aFour summary statistics for kidney function traits were utilized for MR analyses: (from the first row) 1) creatinine-based log-eGFR values of the CKDGen and the UKB data [1], 2) creatinine-based log-eGFR values from the phase 4 CKDGen study [2], 3) cystatin C-based log-eGFR values from CKDGen and UKB [1], and 4) degree of annual eGFR decline including CKDGen and UKB [3].

[1] Stanzick, K. J. *et al.* Discovery and prioritization of variants and genes for kidney function in >1.2 million individuals. *Nat Commun* **12**, 4350 (2021). <https://doi.org/10.1038/s41467-021-24491-0>

[2] Wuttke, M. *et al.* A catalog of genetic loci associated with kidney function from analyses of a million individuals. *Nat Genet* **51**, 957-972 (2019). <https://doi.org/10.1038/s41588-019-0407-x>

[3] Gorski, M. *et al.* Genetic loci and prioritization of genes for kidney function decline derived from a meta-analysis of 62 longitudinal genome-wide association studies. *Kidney Int* **102**, 624-639 (2022). <https://doi.org/10.1016/j.kint.2022.05.021>

Comment R2-22. Please explain how rs57349960 was selected to replace rs7808457 for IL-6.

Response R2-22. Thank you for your comments. Notably, rs7808457 did not replace rs57349960 for IL-6, but there were only two cis-eQTL SNPs for IL-6 available for MR analysis (rs7808457 and rs57349960). One of the two genetic instruments, rs57349960, could not be analyzed because the outcome summary statistics (CKDGen + UKB) did not include the data. Please understand that only the other SNP, rs7808457, could be analyzed for MR analysis (Wald ratio).

Please understand that We have mentioned that rs57349960 was not included in the analysis because it was not included in the outcome summary data in the supplemental table legend to minimize confusion.

In Supplemental table legends;

“^bA cis-pQTL instrument of IL-6 (rs57349960) was not included in the analysis because the summary statistics were unavailable in the outcome GWAS databases.”

Discussion

Comment R2-23. Please discuss the possibility of drug use among participants in outcome GWAS affecting the eGFR measurements, thus affecting the results of the current study.

Response R2-23. Thank you for your valuable comment. We agree that there is a possibility of drug use affecting eGFR measurements in participants with CKDGen or UKB. Certain medications, including angiotensin-converting enzyme inhibitors, angiotensin receptor blockers, and nonsteroidal anti-inflammatory drugs, are commonly used to affect eGFR measurements. However, please note that summary-level MR analysis cannot provide medication-adjusted MR results due to the lack of drug-related data provided by CKDGen. Please understand that this is one of the limitations of summary-level MR analysis.

In the meta-analysis of GWAS from CKDGen and UKB data, each study was adjusted for age, sex, and other covariates and GC-corrected. Additionally, linear mixed model GWAS for log(eGFRcr) and log(eGFRcys) in the UKB included age, age², sex, age × sex, age² × sex, and 20 principal components as covariates [1]. Although these adjustments may not fully modify the effect of medications that affect eGFR measurements in the outcome GWAS, they provide some adjustments for possible confounding factors. These limitations have been addressed and supplemented in the limitations section of this study.

Please note that Because the principal components were adjusted and eGFR is a relatively stable parameter compared to transient variables (e.g., blood pressure), which are immediately affected by related medications, the CKDGen eGFR GWAS meta-analysis has been the most widely used summary-level GWAS to reflect populations' kidney function. Because of the characteristics of eGFR values, the current literature commonly uses CKDGen eGFR GWAS without considering medication history [2-4].

[1] Stanzick, K. J. *et al.* Discovery and prioritization of variants and genes for kidney function in >1.2 million individuals. *Nat Commun* **12**, 4350 (2021).
<https://doi.org/10.1038/s41467-021-24491-0>

[2] Park, S. *et al.* Short or Long Sleep Duration and CKD: A Mendelian Randomization Study. *J Am Soc Nephrol* **31**, 2937-2947 (2020). <https://doi.org/10.1681/asn.2020050666>

[3] Park, S. *et al.* A Mendelian randomization study found causal linkage between telomere attrition and chronic kidney disease. *Kidney Int* **100**, 1063-1070 (2021).
<https://doi.org/10.1016/j.kint.2021.06.041>

[4] Park, S. *et al.* Atrial fibrillation and kidney function: a bidirectional Mendelian randomization study. *Eur Heart J* **42**, 2816-2823 (2021).
<https://doi.org/10.1093/eurheartj/ehab291>

Comment R2-24. The paragraph explaining IL-1 should appear earlier in the discussion, instead of behind the strengths.

Response R2-24. Thank you for the comment. We agree that the paragraph explaining IL-1 should be positioned earlier in the text to provide the sufficient background and its clinical significance for both IL-1 and IL-1ra. In line with this, we revised the manuscript by introducing

IL-1 prior to the paragraph discussing the study's strength in the Discussion section. We believe that the revised manuscript would enhance the organization of our discussion.

Comment R2-25. Please further elaborate what the authors meant by “the MR results of IL-1 alpha were not counterproductive...”.

Response R2-25. Thank you for your comments. The current study found that the genetically predicted serum IL-1ra level was significantly associated with log(eGFR). Please note that As IL-1ra is an antagonist of the IL-1 receptor, we expect that the genetically predicted serum IL-1 level may be inversely associated with log(eGFR). However, the current study could not support the inverse association between serum IL-1 α and kidney function. Please note that this result may be due to the weak-instrument effect, as we mentioned earlier in Response R2-8. However, we agree that the current sentence “the MR results of IL-1 α were not counterproductive” could be debatable. In the revised manuscript, we have further elaborated as below.

In Discussion;

“Our study has some limitations. First, the number of genetic instruments was insufficient for some ILs, which restricted the type of sensitivity analysis that could be performed. Although the strengths of the instruments were evaluated using F-statistics and r^2 [1], there were weak-powered instruments with F-statistics <10 . It should be noted that there is potential for false-negative bias, particularly when the true effect is below the minimum effect size (Supplemental Table 6). Thus, one may not preclude the possibility of a kidney function effect based on the null findings in this study. Specifically, we could expect that the IL-1 α , which agonizes the IL-1 receptor, may produce an inverse effect on kidney function. However, the current study could not support the inverse association between serum IL-1 α and kidney function. This may be explained by the broader biological function of IL-1ra than that of IL-1 α , including the blockade of IL-1 receptors from binding with IL-1 agonists [2]. However, the difference in the statistical power may have caused the discrepancy. Second, the effect estimates of MR analysis are for lifetime exposure to genetic predisposition and does not consider temporal and spatial fluctuations in gene expression in tissues. Additionally, as the overall effect sizes in the results were small, a transient change with a higher degree of ILs may have different effects on kidney function. Third, the generalizability is limited as the analysis was mainly based on European ancestry samples.”

[1] Pierce BL, Ahsan H, Vanderweele TJ. Power and instrument strength requirements for Mendelian randomization studies using multiple genetic variants. Int J Epidemiol. 2011

[2] Netea, M. G., van de Veerdonk, F. L., van der Meer, J. W., Dinarello, C. A. & Joosten, L. A. Inflammation-independent regulation of IL-1-family cytokines. *Annu Rev Immunol* **33**, 49-77 (2015). <https://doi.org/10.1146/annurev-immunol-032414-112306>

Comment R2-26. Your result shown that horizontal pleiotropy was not evident through MR-Egger intercept. Why still list this as a limitation?

Response R2-26. Thank you for the comment. We agree that the horizontal pleiotropy was not evident in MR-Egger intercept. In the revised manuscript, we have excluded the statement regarding horizontal pleiotropy in the limitation in Discussion section.

Reviewer: 3

Remarks to the Author

Comment R3-1. The CKDGen database contains CKD, serum creatinine, uric acid protein, proteinuria and other data. Why did the authors choose only eGFR to measure renal function? And did the data used by the authors come from data from which year in the database? This should be explicitly described in the text.

Response R3-1. Thank you for the thoughtful comment. We agree with your comment that eGFR is not the sole parameter of kidney function, and CKD, uric acid, proteinuria and other data would be an important kidney function parameter that should have been considered because it is independently associated with health outcomes from eGFR.

However, please understand that the current guideline in the nephrology field, including the most respected KDIGO guideline (URL: <https://kdigo.org/guidelines/ckd-evaluation-and-management/>), the degree and present of kidney function impairment is majorly determined by eGFR and urine albuminuria. If we refer to the guideline,

“1.2: STAGING OF CKD

1.2.1: We recommend that CKD is classified based on cause, GFR category, and albuminuria category (CGA). (1B)”

The usage of eGFR as the initial assessment step is also described in the guideline.

“1.4.3: Evaluation of GFR

1.4.3.1: We recommend using serum creatinine and a GFR estimating equation for initial assessment. (1A)

1.4.3.2: We suggest using additional tests (such as cystatin C or a clearance measurement) for confirmatory testing in specific circumstances when eGFR based on serum creatinine is less accurate. (2B)”

Current medicine is very dependent on eGFR values to assess kidney function because there is little evidence that indicates an impairment of kidney function in those with preserved GFR. In our opinion, GFR does reflect the possibility of other functional impairments related to the kidney and thus should be the primary marker to be assessed for its causality. This can also be supported in the literature, as most Mendelian randomization studies assess eGFR or eGFR-based outcomes and genetically predicted eGFR values, rather than the diverse biological effects that you mentioned, as other biological consequences would fundamentally result from GFR decline and this can be supported by many references [1-5]. We believe that assessing other phenotypes and addressing the phenotype that would represent kidney function would be highly likely to raise debates compared to other Mendelian randomization studies focusing on eGFR.

In addition, it would be less debatable whether eGFR is the most readily available kidney function marker is eGFR, including the representative kidney function marker as the main

outcome in the current study has clinical value. Thus, our study suggests a protective effect of IL-1ra on kidney function and that the IL-1 inhibitor could be safely administered in patients with reduced eGFR. Therefore, including eGFR as the outcome of interest would have particular clinical value.

Regarding the year from which the dataset is provided, please note that we conducted MR analysis using four different datasets, including two meta-analyses of log(eGFRcr), one from CKDGen + UKB meta-analysis by Stanzick et al. (2021) and the other from CKDGen meta-analysis by Wuttke et al. (2019), a meta-analysis of log(eGFRcys) of CKDGen + UKB by Stanzick et al. (2021), and a meta-analysis of annual eGFRcr decline of CKDGen + UKB by Gorski et al. (2021).

In summary, we used the summary statistics of log(eGFR) and the degree of eGFR decline, as eGFR values are currently the standard parameter for assessing kidney function for determining various medical practices. We have provided evidence for choosing eGFR for the outcome of kidney function traits and the year from which the dataset originated in the revised manuscript, as follows.

In our Methods;

“Kidney function outcome in the MR analysis

We used three eGFR outcome datasets as eGFR values are a current standard parameter for the assessment of kidney function; 1) the meta-analysis for creatinine-based log-eGFR values of the CKDGen and UKB data from 2021 (n = 765,348) which has strength as this is the currently largest GWAS meta-analysis for eGFR trait [6], 2) creatinine-based log-eGFR values from the phase 4 CKDGen study from 2019 (n = 435,581) [7],¹ and 3) cystatin C-based log-eGFR values from the CKDGen and UKB (n = 460,826) from 2021 [6], which has particular strength in that the dataset can be used to test the causal estimates towards cystatin C-based eGFR values which is less affected from non-kidney factors than creatinine-based levels [8]. The main analysis was performed using the largest GWAS meta-analysis of creatinine-based log-eGFR from CKDGen and UKB because of the greater statistical power and lower possibility of weak-instrumental bias. Nevertheless, considering the potential for healthy volunteer bias resulting from the relatively lower prevalence of CKD in the UKB dataset [9], replication analyses employing the other two GWAS meta-analysis datasets were conducted. Additionally, the UKB dataset was independent of the CKDGen data, fulfilling an independent replication analysis, and was independent of the samples included in the GWAS for genetic instrument development. Such a two-sample MR without participant overlap has statistical power even if the results are affected by instrumental power because the potential bias would be towards false-negative findings. Therefore, a positive finding from such an independent two-sample MR is more likely to reflect true causal effects [10].

Furthermore, to test whether IL levels have causal effects on accelerated eGFR decline rather than on a static value, we implemented MR analysis using a GWAS meta-analysis of

the degree of annual eGFR decline from CKDGen and UKB from 2022 (n = 343,339) as another outcome dataset [11].”

[1] Ellervik C, Mora S, Ridker PM, Chasman DI. Hypothyroidism and Kidney Function: A Mendelian Randomization Study. *Thyroid*. 2020;30(3):365-379. doi:10.1089/thy.2019.0167

[2] Kennedy OJ, Pirastu N, Poole R, et al. Coffee Consumption and Kidney Function: A Mendelian Randomization Study. *Am J Kidney Dis*. 2020;75(5):753-761. doi:10.1053/j.ajkd.2019.08.025

[3] Park S, Lee S, Kim Y, Lee Y, Kang MW, Kim K, Kim YC, Han SS, Lee H, Lee JP, Joo KW, Lim CS, Kim YS, Kim DK. Short or Long Sleep Duration and CKD: A Mendelian Randomization Study. *J Am Soc Nephrol*. 2020 Dec;31(12):2937-2947. doi:10.1681/ASN.2020050666. Epub 2020 Oct 1. PMID: 33004418; PMCID: PMC7790216.

[4] Park S, Lee S, Kim Y, et al. Causal effects of relative fat, protein, and carbohydrate intake on chronic kidney disease: a Mendelian randomization study [published online ahead of print, 2021 Feb 10]. *Am J Clin Nutr*. 2021;nqaa379. doi:10.1093/ajcn/nqaa379

[5] Zhao JV, Schooling CM. Sex-specific associations of insulin resistance with chronic kidney disease and kidney function: a bi-directional Mendelian randomisation study. *Diabetologia*. 2020;63(8):1554-1563. doi:10.1007/s00125-020-05163-y

[6] Stanzick, K. J. *et al.* Discovery and prioritization of variants and genes for kidney function in >1.2 million individuals. *Nat Commun* **12**, 4350 (2021). <https://doi.org/10.1038/s41467-021-24491-0>

[7] Wuttke, M. *et al.* A catalog of genetic loci associated with kidney function from analyses of a million individuals. *Nat Genet* **51**, 957-972 (2019). <https://doi.org/10.1038/s41588-019-0407-x>

[8] Lees, J. S. *et al.* Glomerular filtration rate by differing measures, albuminuria and prediction of cardiovascular disease, mortality and end-stage kidney disease. *Nat Med* **25**, 1753-1760 (2019). <https://doi.org/10.1038/s41591-019-0627-8>

[9] Bycroft, C. *et al.* The UK Biobank resource with deep phenotyping and genomic data. *Nature* **562**, 203-209 (2018). <https://doi.org/10.1038/s41586-018-0579-z>

[10] Burgess, S., Davies, N. M. & Thompson, S. G. Bias due to participant overlap in two-sample Mendelian randomization. *Genet Epidemiol* **40**, 597-608 (2016). <https://doi.org/10.1002/gepi.21998>

[11] Gorski, M. *et al.* Genetic loci and prioritization of genes for kidney function decline derived from a meta-analysis of 62 longitudinal genome-wide association studies. *Kidney Int* **102**, 624-639 (2022). <https://doi.org/10.1016/j.kint.2022.05.021>

Comment R3-2. What are the specific screening criteria for instrumental variables? What is their F-value?

Response R3-2. Thank you for your comments. We agree that our specific screening criteria for the selection of instrumental variables (IVs) based on their power should be mentioned in the manuscript. We primarily set the criteria for F-statistics ≥ 10 or $r^2 \geq 0.01$, based on the current guideline of performing Mendelian randomization, which states that instrumental variants with an F-statistic < 10 are often labeled as weak instruments [1]. However, we are aware that some of our instruments do not meet these criteria, as shown in Supplemental Table 2.

We understand the limitations of the weak instruments, as stated in the limitations section. However, please note that our primary focus was the association between genetically predicted serum IL-1ra levels and kidney function, in which most of the genetic instruments had sufficient power, with F-statistics ≥ 10 . In addition, we aimed to conserve SNPs whenever possible so as not to exclude significant signals in the MR analysis.

Please understand that we primarily considered IVs with F-statistics ≥ 10 or $r^2 \geq 0.01$ as reliable instruments; however, we decided to conserve the SNPs despite them not satisfying the threshold with awareness of the limitation, including false-negative findings of weak instruments.

[1] Burgess S, Davey Smith G, Davies NM, et al. Guidelines for performing Mendelian randomization investigations. Wellcome Open Res. 2020;4:186. Published 2020 Apr 28. doi:10.12688/wellcomeopenres.15555.2

Comment R3-3. Post hoc power for univariable primary analyses should be calculated to estimate the minimum effect size.

Response R3-3. Thank you for your comments. We agree that a post-hoc power analysis can provide information about the minimum effect size. However, please understand that the post hoc power could not be calculated in the current study because all outcome data analyzed in our study did not provide a standard deviation (SD) value, which is essential for calculating post hoc power. Instead, we calculated the minimum effect size of ILs for a power of 80% using the known information, including outcome sample size (the meta-analysis for log(eGFR_{cr}) of the CKDGen and UKB, N = 765,348) and r^2 of each IL. The effect size was calculated using the most commonly used application for calculating post hoc power (<https://shiny.cnsgenomics.com/mRnd/>) [2]. The type-I error rate is set to 0.25.

In other words, the set of instrumental variables powered less than the minimum effect size might not have been significant in this study. Please understand that we needed to calculate the minimum effect sizes and provide the possibility of insignificant findings in instrumental

variables that have lower power than the minimum effect size because of the absence of SD information.

In the revised manuscript, the minimum effect size calculation is described in the Methods section and provided in Supplemental Table 6. Regarding weak-powered instruments, please note that we have elaborated on the possibility of false positives in the limitations of the Discussion section.

In Methods

“Other statistical aspects of the MR analysis

The minimum effect size of each IL for a power of 80% was calculated in the application for calculating the post hoc power (<https://shiny.cnsgenomics.com/mRnd/>) using the meta-analysis dataset for log-eGFRcr of the CKDGen and UKB datasets and r^2 values of the instrumental variables of each IL at a type-I error rate of 0.25.”

In Discussion

“Our study has some limitations. First, the number of genetic instruments was insufficient for some ILs, which restricted the type of sensitivity analysis that could be performed. Although the strengths of the instruments were evaluated using F-statistics and r^2 [1], there were weak-powered instruments with F-statistics <10. It should be noted that there is potential for false-negative bias, particularly when the true effect is below the minimum effect size (Supplemental Table 6). Thus, one may not preclude the possibility of a kidney function effect based on the null findings in this study.”

Supplemental Table 6. The minimum effect size for 80% power in each IL

Interleukin	pQTL		eQTL		pQTL and eQTL	
	r^2	Minimum effect size	r^2	Minimum effect size	r^2	Minimum effect size
IL-1 α	–	–	0.003	0.041	0.003	0.041
IL-1ra	0.075	0.008	0.017	0.017	0.092	0.007
IL-2ra	0.260	0.02	0.130	0.028	0.390	0.003
IL-6	0.002	0.051	0.001	0.071	0.003	0.041
IL-7	0.005	0.032	–	–	0.005	0.032
IL-8	0.004	0.036	0.005	0.032	0.009	0.024

IL-12p70	0.002	0.051	–	–	0.002	0.051
IL-16	0.037	0.012	0.031	0.013	0.068	0.008
IL-18	0.051	0.01	0.024	0.015	0.075	0.008

IL = interleukin, QTL = quantitative trait loci

The effect size was calculated at <https://shiny.cns.genomics.com/mRnd/>. The CKDGen + UKB meta-analysis of creatinine-based log-eGFR values was used as the outcome dataset, and the type-I error rate was set at 0.25.

[1] Pierce, B. L., Ahsan, H. & Vanderweele, T. J. Power and instrument strength requirements for Mendelian randomization studies using multiple genetic variants. *Int J Epidemiol* **40**, 740-752 (2011). <https://doi.org/10.1093/ije/dyq151>

[2] Burgess S, Davey Smith G, Davies NM, et al. Guidelines for performing Mendelian randomization investigations. *Wellcome Open Res.* 2020;4:186. Published 2020 Apr 28. doi:10.12688/wellcomeopenres.15555.2

Comment R3-4. If only 1 instrument snp is available, the Wald ratio should be used.

Response R3-4. Thank you for your comments. Please note that We calculated the coefficient ratio, also termed the Wald ratio, to generate MR estimates with a single instrumental variant. In the current study, we analyzed ILs that comprised only one SNP using the Wald ratio, but represented the method as a “ratio of coefficient.” We agree that referring to the Wald ratio as the “ratio of coefficient” may cause confusion for our readers. To address this issue, we provided explanatory statements in the Methods section.

In our Methods;

“Two-sample MR analysis with summary-level data

Summary-level MR analysis was performed using two sets of genetic instruments (cis-pQTLs and cis-eQTLs) for each IL on the four outcome datasets. First, we implemented a multiplicative random-effect inverse variance weighted (IVW) method when more than one genetic instrument was available to construct an instrumental variable for a given IL. Provided with only a single genetic instrument, the ratio of coefficient, also referred to as the Wald ratio, was applied to measure MR estimates [1]”

We believe that this explanation clarifies the terminology and avoids potential misunderstandings. If you consider a different description to be more appropriate, please let us know and we will change the terminology.

[1] Burgess, S., Butterworth, A. & Thompson, S. G. Mendelian randomization analysis with multiple genetic variants using summarized data. *Genet Epidemiol* **37**, 658-665 (2013). <https://doi.org:10.1002/gepi.21758>.

Comment R3-5. There is a potential association between different IL-ras, and the authors should perform multivariable Mendelian randomization simultaneously to exclude potential effects.

Response R3-5. Thank you for your comments. Please understand that We could not perform multivariable MR imaging in this study because of the lack of full summary-level data. This has been pointed out in the Limitations section of the Discussion section. Furthermore, the effect of the current study will remain independent of the significance of multivariable MR results. Even if the adjusted MR results showed non-significant findings, they would suggest the presence of vertical pleiotropy. We believe that the main message of this study remains intact, regardless of multivariable MR analysis.

REVIEWERS' COMMENTS:

Reviewer #1 (Remarks to the Author):

I appreciate the clarification provided by the authors. Their responses to all three of my questions were excellent, and they have taken the initiative to revise the "Discussion" section based on my third comment. I want to express my satisfaction with their thoroughness and attention to detail. At this point, I have no further questions or comments.

Reviewer #2 (Remarks to the Author):

I commend the authors for their effort in addressing the comments. I have only several minor comments, which are stated below:

- line 147: datasets cannot be implemented. Do the authors mean utilised (or anything similar)?
- I am confused by the concept of replication in the manuscript. The authors repeatedly stated that involving UKB data counts as an independent replication of the analysis (line 51-52, 148-149, 162-163), which made me expecting an MR involving outcome data from UKB only. However, UKB data was always meta-analysed together with CKDGen data in this work, making the authors' description inaccurate and needs clarification.
- On the same note, the authors claimed the analysis involving eGFR-cys was a replication of analysis involving eGFRcr, which to me is confusing. I suggest either state this as a validation or explicitly state that the two eGFR indices are essentially identical.

Reviewer #3 (Remarks to the Author):

My concerns have been solved.

Response Letter

Editorial Board: 1

In light of their advice we are delighted to say that we are happy, in principle, to publish a suitably revised version in Communications Biology under the open access CC BY license (Creative Commons Attribution v4.0 International License).

We therefore invite you to revise your paper one last time to address the remaining concerns of our reviewers.

Response: Thank you for the valuable comment. We are very glad to hear that our revised version aligns with the high standards of your esteemed journal. The responses to the minor revisions suggested by reviewers are described below.

Reviewer: 1

I appreciate the clarification provided by the authors. Their responses to all three of my questions were excellent, and they have taken the initiative to revise the "Discussion" section based on my third comment. I want to express my satisfaction with their thoroughness and attention to detail. At this point, I have no further questions or comments.

Response 1: We truly appreciate your comment in the previous revision. Your comments were highly valued and helped our work to provide more comprehensive and detailed explanations. Thus, we are delighted to hear our responses to your questions succeeded in satisfying your concerns. If you have any further comments, please do not hesitate to reach out. Once again, thank you for the valuable opinions.

Reviewer: 2

I commend the authors for their effort in addressing the comments. I have only several minor comments, which are stated below:

Comment 2-1: line 147: datasets cannot be implemented. Do the authors mean utilised (or anything similar)?

Response 2-1: Thank you for the comment. We agree that the word “implemented” should be changed, so we have amended the sentence as below.

In our Method;

“Kidney function outcome in the MR analysis

We used two independent datasets, the CKDGen consortium data and UKB, as outcome datasets for log-transformed eGFR values because validation analysis is crucial for the validity of a genetic study.”

Comment 2-2: I am confused by the concept of replication in the manuscript. The authors repeatedly stated that involving UKB data counts as an independent replication of the analysis (line 51-52, 148-149, 162-163), which made me expecting an MR involving outcome data from UKB only. However, UKB data was always meta-analysed together with CKDGen data in this work, making the authors' description inaccurate and needs clarification.

Response 2-2: Thank you for the valuable comment. In the current study, there are two sets of GWAS meta-analysis of log-transformed creatinine based eGFR: 1) UKB + CKDGen, and 2) CKDGen. We performed main analysis using UKB + CKDGen meta-analysis, and found the result was also maintained in the CKDGen meta-analysis dataset. As you mentioned, we agree that term “replication” can be confusing for some readers in that it brings up expectations for other analyses such as analyzing UKB data only. In agreement with your concerns, we clarified that the meta-analysis of log-eGFRcr from CKDGen was the validation of the main analysis in the revised manuscript. Specifically, we have presented that the meta-analysis of log-eGFRcr from CKDGen was the “validation” study in the figure 1 and clarified it in the figure legend also.

In the Figure 3 and figure legend;

“Figure 3. Study flow diagram. The study included summary-level Mendelian randomization analysis testing the causal effects between circulating interleukin (IL) level and kidney function. The cis-genetic instruments for genetically predicted serum IL levels, including cis-pQTL and cis-eQTL instruments, were developed from the previous genome-wide association study (GWAS) meta-analysis including 38,424 participants. Among 376 SNPs, 80 SNPs of nine ILs were utilized for the Mendelian randomization analysis. The summary statistics for kidney function traits were obtained from four respective GWAS meta-analysis datasets: the meta-analysis for creatinine-based log-eGFR values of CKDGen and UKB (n = 1,201,930), creatinine-based log-eGFR values from the phase 4 CKDGen study (n = 765,348), cystatin C-based log-eGFR values from CKDGen and UKB (n = 460,826), and degree of annual eGFR decline from CKDGen and UKB (n = 343,339). Below the boxes of each dataset, it is indicated whether the datasets were analyzed as “Main analysis” or “Validation”. UKB = UK Biobank, SNP = single nucleotide polymorphism, GWAS = genome-wide association study, eGFRcr = estimated glomerular filtration rate by creatinine, eGFRcys = estimated glomerular filtration rate by cystatin C”

Comment 2-3: On the same note, the authors claimed the analysis involving eGFR-cys was a replication of analysis involving eGFRcr, which to me is confusing. I suggest either state this as a validation or explicitly state that the two eGFR indices are essentially identical.

Response 2-3: Thank you for your comment. Please note that the GFR estimates calculated from creatinine and cystatin-C are not equal, as they are derived from similar

but distinct equations. However, in this study, our focus is on considering both values as representative of kidney function in terms of estimating GFR and possessing comparable clinical implications. Still, we agree with your observation that the term “replication” might be inappropriate, as the two eGFR indices are not identical. In the revised manuscript, we have clarified that the analyses using the cystatin C-meta-analysis data constitute a “validation” rather than a replication analysis as below. We believe we have adequately addressed the issues and concerns raised in the comment. However, if you have any further comments, please inform us.

In our Method;

“Kidney function outcome in the MR analysis

Nevertheless, considering the potential for healthy volunteer bias resulting from the relatively lower prevalence of CKD in the UKB dataset [1], validation analyses employing the other two GWAS meta-analysis datasets were conducted. Additionally, the UKB dataset was independent of the CKDGen data, fulfilling an independent validation analysis, and was independent of the samples included in the GWAS for genetic instrument development.”

[1] Bycroft, C. *et al.* The UK Biobank resource with deep phenotyping and genomic data. *Nature* **562**, 203-209 (2018). <https://doi.org:10.1038/s41586-018-0579-z>

Reviewer: 3

My concerns have been solved.

Response 3: Thank you for the valuable opinions you've provided in the previous revision. Through your response, we could escalate our paper's quality.